# Social and behavioral factors related to blood pressure measurement: A cross-sectional study in Bhutan

**Hiromi Kohori Segawa**[1,2], **Hironori Uematsu**[1], **Nidup Dorji**[3], **Ugyen Wangdi**[3], **Chencho Dorjee**[3], **Pemba Yangchen**[4], **Susumu Kunisawa**[1], **Ryota Sakamoto**[5], **Yuichi Imanaka**[1]*

**1** Department of Healthcare Economics and Quality Management, Graduate School of Medicine, Kyoto University, Kyoto City, Kyoto, Japan, **2** Kokoro Research Center, Kyoto University, Kyoto City, Kyoto, Japan, **3** Faculty of Nursing and Public Health, Khesar Gyalpo University of Medical Sciences of Bhutan, Thimphu, Kingdom of Bhutan, **4** Non-Communicable Diseases Division, Ministry of Health in Bhutan, Thimphu, Kingdom of Bhutan, **5** Centre for Southeast Asian Studies, Kyoto University, Kyoto City, Kyoto, Japan

* imanaka-y@umin.net

**Data Availability Statement:** Since the Ministry of Health, Bhutan is the authority for the data, the authors were not able to make it available with this paper. However, the original report (World Health

## Abstract

Cardiovascular disease is a leading cause of death in the Kingdom of Bhutan, and early detection of hypertension is critical for preventing cardiovascular disease. However, health-seeking behavior, including blood pressure measurement, is infrequently investigated in Bhutan. Therefore, this study investigated factors related to blood pressure measurement in Bhutan. We performed a secondary data analysis of a target population of 1,962 individuals using data from the "2014 Bhutan STEPS survey data"as a cross-sectional study. Approximately 26% of those with hypertension who were detected during the STEPS survey had never had their blood pressure measured. Previous blood pressure measurement was significantly associated with age and working status in men (self-employed [odds ratio (OR): 0.219, 95% CI: 0.133–0.361], non-working [OR: 0.114, 95% CI: 0.050–0.263], employee [OR: 1.000]). Previous blood pressure measurement was significantly associated with higher income in women (Quartile-2 [OR: 1.984, 95% CI: 1.209–3.255], Quartile-1 [OR: 2.161, 95% CI: 1.415–3.299], Quartile-4 [OR: 1.000]). A family history of hypertension (OR: 2.019, 95% CI: 1.549–2.243) increased the likelihood of having experienced a blood pressure measurement in both men and women. Multivariate logistic regression showed that people with unhealthy lifestyles (high salt intake [adjusted odds ratio (AOR): 0.247, 95% confidence interval (CI): 0.068–0.893], tobacco use [AOR: 0.538, 95% CI: 0.380–0.761]) had a decreased likelihood of previous blood pressure measurement. To promote the early detection of hypertension in Bhutan, we suggest that more attention be paid to low-income women, non-working, self-employed, and low-income men, and a reduction of barriers to blood pressure measurement. Before the STEPS survey, a substantial number of hypertensive people had never had their blood pressure measured or were unconcerned about their health. As a result, we propose that early blood pressure monitoring and treatment for people with hypertension or at higher risk of hypertension be given increased emphasis.

Organization and Ministry of Health in Bhutan. National Survey for Noncommunicable Disease Risk Factors and Mental Health using WHO STEPS Approach in Bhutan – 2014) is available from https://apps.who.int/iris/handle/10665/204659. Requests to access the raw data can be directed to the Research Ethics Board of Health at the Ministry of Health, Royal Government of Bhutan. Contact: P. O. Box: 726, Kawajangsa, Thimphu, Bhutan Phone No: +975-2-328095, 321842, 322602, 328091.

**Funding:** This study was supported by K. Matsushita Foundation (http://matsushita-konosuke-zaidan.or.jp/en/) in the form of a grant (19-048) to HKS. This study was also supported by Kyoto University (https://ipcr.cseas.kyoto-u.ac.jp/), Program of Collaborative Research at the Centre for Southeast Asian Studies (IPCR-CSEAS) with funding (type 4) for YI and funding (type 7) for HKS. Kyoto University also supported this study with funding (Ishizue 2022) for YI. There are no grant numbers associate with the support from Kyoto University. This study was also supported by Japan Society for the Promotion of Science (JSPS Kakenhi) in the form of a grant (JP19H01075) to YI. The funders played no role in the study design, data collection and analysis, decision to publish, or preparation of the manuscript.

**Competing interests:** The authors have declared that no competing interests exist. The funders played no role in the study design, data collection and analysis, decision to publish, or preparation of the manuscript.

## Introduction

According to the World Health Organization (WHO), 38 million people die from non-communicable diseases (NCDs) annually. Approximately three-quarters of these deaths (28 million) occur in low- and middle-income countries [1]. This increase in NCDs, such as cardiovascular disease, has led to an economic burden on individuals, families, and society [2].

Hypertension is one of the strongest risk factors for almost all cardiovascular diseases. The course of cardiovascular disease is indicated by asymptomatic alterations in numerous organs linked with hypertension [3]. Secondary prevention strategies such as screening, early detection, counseling, and continued follow-up of people with high blood pressure are essential to prevent cardiovascular-related diseases. There is, however, a distinction between the true prevalence of hypertension and the rates of diagnosis and treatment [4].

Similarly, in the Kingdom of Bhutan, mortality due to NCDs has increased from 53% (2008) to 69% (2016) [5, 6] and is the leading cause of mortality [6]. The government of Bhutan provides free access to basic public health services for all its citizens, and the Ministry of Health has initiated primary prevention programs such as maintaining a healthy diet, avoiding alcohol and tobacco, and exercising, all of which are essential in preventing NCDs [7–9].

However, a previous qualitative study indicated that most Bhutanese have inadequate knowledge about how to protect themselves from hypertension through daily practices, although they wish to be healthy because health is an important factor for happiness [10]. The participants were diagnosed with hypertension during previous fieldwork conducted in 2017, and some had never had their blood pressure measurement before this fieldwork and were unaware of the importance of regular blood pressure measurement [10]. Furthermore, according to the STEPS survey report, "Around 31.3% of the study population had never had their blood pressure checkup. The prevalence of raised blood pressure or hypertension (SBP$\geq$140 and/or DBP$\geq$90) was 35.7% (men 35.5%, women 35.9%) when those currently using medication were included." [7].

Blood pressure measurement is vital as a first step in subsequent hypertension prevention. Therefore, the STEPS survey data can be used to determine which social backgrounds are associated with lower blood pressure measurements in Bhutan. This can also reveal which high-risk behaviors for hypertension are associated with blood pressure measurement. However, in Bhutan, these topics have received less attention, especially in terms of quantitative and representative research.

In this study, we looked into the social backgrounds of those who had less access to blood pressure measurement, as well as behavioral factors associated with less blood pressure measurement experience, in order to contribute to more hypertension prevention efforts in Bhutan.

We need to consider both aspects of " access and behaviors" and "vulnerability" when examining the prevention of severe hypertension and early detection of hypertension. We have a related manuscript, the aspect from "vulnerability". https://doi.org/10.1371/journal.pone.0256811.

### Objectives

To use the STEPS survey in Bhutan to investigate

1. The social factors that are associated with less blood pressure measurement experience.

2. Whether high-risk behavioral factors for hypertension are associated with less blood pressure measurement experience.

## Methods

### Study design

This cross-sectional study was conducted using data from the "National Survey for Noncommunicable Disease Risk Factors and Mental Health using WHO STEPS Approach in Bhutan– 2014" [7].

### Study setting

Bhutan is located in the eastern Himalayas between India in the east, west, and south and China in the north. This region has an area of 38,394 km$^2$. As of 2020, the Bhutanese population was estimated to be 748,931 [11]. The gross domestic product (GDP) per capita was estimated to be USD 3,411.94 in 2019 [11]. Bhutan's national development is based on the philosophy of "gross national happiness (GNH)", which aspires to sustainable development and happiness for all its citizens [12, 13].

### Data sources

Data were derived from the "National Survey for Noncommunicable Disease Risk Factors and Mental Health using WHO STEPS Approach in Bhutan– 2014", which was conducted by WHO and the government of Bhutan from March to June 2014 [7]. The sample size was 2,912, which was considered sufficient to represent the target population (adults aged 18–69 years) in Bhutan (S1 File). To ensure the representativeness of the sample, the study applied multistage cluster sampling combined with probability measures proportionate to the size and systematic random sampling. The strata level considered was urbanicity (rural: urban = 7:3); an area block was designated as the cluster level (n = 182) and was selected from *gewogs* (group of villages). The Kish method was applied to select participants from each household using age and gender as variables, and the sampling framework from the "Population and Housing Census of Bhutan 2005" was employed [14]. According to the sampling plan, trained health staff recruited participants in each sampling field where the participants were living. Trained health staff collected data through face-to-face interviews. The number of valid respondents was 2,822 with a response rate of 97%. A previous report by WHO and the Ministry of Health in Bhutan provides the details of the survey procedure [7].

### Definitions of variables

**Objective variables.** Previous blood pressure measurement was defined as 1 for "Yes" and 0 for "No" for the questionnaire item in the survey "Have you ever had your blood pressure measured by a doctor or other health worker?" (This question regarded the period before the survey, excluding blood pressure measurement that occurred during the survey).

**Explanatory variables.** Explanatory variables were selected and categorized mainly by following the WHO guidelines [15] and previous literature [16–30].

Gender (men, women), marital status (married or cohabiting, never married, separated or divorced or widowed), age, level of education (non-formal education, elementary education [1–10 years], high school education [11–12 years], and tertiary education [above 12 years]; a promotion test is given in the 10th and 12th years in Bhutan), working status (i.e., employed, self-employed, and non-working), residential area (urbanicity), income (Quartile-1 (Q1) (Nu. 60,001+: USD 805.8+), Quartile-2 (Q2) (Nu. 30,001–60,000: USD 403.0–805.7), Quartile-3 (Q3) (Nu. 9,001–30,000: USD 120.9–402.9), Quartile-4 (Q4) (Nu. 0–9,000: USD 0–120.8)), and survey language (Dzongkha, Tshanglakha, Lhotshamkha, English) were also recorded.

Tobacco use (never, current, or any [not only smoking but also chewing tobacco]) [15], alcohol consumption (none, light or moderate, or heavy) [15, 21], fruit and vegetables consumption (<5, 5, or >5 portions per day) [15, 19], physical activity level (≥150 min of moderate activity per week or <150 min of moderate activity per week) [15, 20], and estimated salt intake (<5 g, 5 g, or >5 g per day with a cutoff point determined according to the WHO guidelines and the Tanaka formula used for estimation: Salt intake per day(g) = (21.98 × ((spot urinary sodium/ (spot urinary creatinine × 10/0.0884)) × (((14.89 × weight [Kg]) +(16.14 × height [m]) × (2.04 × age [years])) - 2244.45)) × 0.392)/17.1) were investigated [15, 22–26]. All variables were measured using the WHO guidelines.

Hypertension was defined by the following criteria: (1) systolic blood pressure (SBP) ≥140 mmHg or diastolic blood pressure (DBP) ≥90 mmHg measured as the mean of three measurements taken by health staff in the STEPS survey; (2) a previous diagnosis of hypertension by healthcare workers; (3) current treatment with medication for hypertension [15].

## Statistical methods

Descriptive statistics were used to describe the characteristics of the study population. Univariable logistic regression was performed to evaluate the associations between objective variables and each explanatory variable. All available and selected data from the STEPS survey were considered to be related to the objective variables from previous literature. Among those variables that may be related to blood pressure measurement, we selected explanatory variables to be fed into the model using a directed acyclic graph (S2 File).

Health behavior such as blood pressure measurement is affected by various factors. In particular, we considered both individual and social environmental factors as targets for health promotion interventions. "Interpersonal" processes are known to shape human behavior and lifestyles [25, 26].

Health policy is common for all Bhutanese across the country. Some but not all of the primary prevention programs for NCDs, such as displays of posters in health facilities, had already started by 2014 at the time of the survey. The number of primary health centers (PHCs) was also increasing in rural areas due to the expansion of primary health care. We predict that the situations of diagnosis and risk perception were also diverse. Furthermore, risk perception (threat) is also known to affect human behavior [3]. We selected individual and social environmental factors that influence interpersonal decisions or close networks. Gender, marital status, age, level of education, working status, income, and survey language were among the selected characteristics. Tobacco use, alcohol consumption, fruit and vegetable consumption, physical activity level, estimated salt intake, history of hypertension, and family history of hypertension were all chosen as lifestyle-related and biological risk perception factors.

We evaluated the association between previous blood pressure measurement and the individual and social environmental factors affecting interpersonal decisions or close networks through descriptive and univariable logistic regression, the association between previous blood pressure measurement and the high-risk behavioral factors of hypertension through descriptive and multivariable logistic regression (S2 File: Directed acyclic graph for multivariable logistic regression). Through the directed acyclic graph, we defined the minimal sufficient adjustment sets for estimating the total effect of high-risk behavioral factors of hypertension on the experience of having a blood pressure measurement.

All data were analyzed using IBM Statistical Package for Social Sciences version 23 with the module for Complex Sample Analysis (IBM Corp., Armonk, NY, USA), excluding the nonweighted calculation for Table 1 (mean and standard deviation). All estimates are presented with 95% confidence intervals (CIs) after adjusting for weight following the WHO guidelines

**Table 1. Distribution of socio-demographic characteristics and life behaviors.**

| | | n | Mean[1] | Standard deviation[1] | Weight adjusted[2] | BP measured[3] | Estimated[4] | 95% CI | | |
|---|---|---|---|---|---|---|---|---|---|---|
| Total | | 1962 | | | 100% | 1402 | 67.1% | ( 62.9% | – | 71.2% ) |
| Gender | Men | 772 | | | 57.6% | 486 | 60.8% | ( 54.5% | – | 66.8% ) |
| | Women | 1190 | | | 42.4% | 916 | 75.8% | ( 72.4% | – | 78.9% ) |
| Marital status | Married or cohabiting | 1588 | | | 83.1% | 1158 | 69.6% | ( 64.9% | – | 73.8% ) |
| | Never married | 155 | | | 10.4% | 86 | 46.7% | ( 37.2% | – | 56.4% ) |
| | Separated, divorced, or widowed | 219 | | | 6.5% | 158 | 69.1% | ( 59.9% | – | 77.0% ) |
| Age | | | 40.25 | 12.36 | | | | | | |
| | 18–29 years | 443 | | | 28.5% | 297 | 61.4% | ( 54.7% | – | 67.7% ) |
| | 30–39 years | 571 | | | 36.6% | 427 | 67.7% | ( 61.3% | – | 73.6% ) |
| | 40–49 years | 475 | | | 17.1% | 349 | 73.3% | ( 67.9% | – | 78.2% ) |
| | 50–59 years | 306 | | | 11.4% | 213 | 68.6% | ( 61.2% | – | 75.2% ) |
| | 60–69 years | 167 | | | 6.3% | 116 | 70.0% | ( 60.3% | – | 78.2% ) |
| Education | | | 4.90 | 12.05 | | | | | | |
| | No formal education | 1203 | | | 54.1% | 828 | 63.6% | ( 58.4% | – | 68.5% ) |
| | 1–10 years | 600 | | | 35.9% | 453 | 71.2% | ( 65.8% | – | 76.0% ) |
| | 11–12 years | 99 | | | 6.2% | 70 | 63.0% | ( 49.0% | – | 75.1% ) |
| | Above 12 years | 60 | | | 3.7% | 51 | 86.5% | ( 72.6% | – | 94.0% ) |
| Working status | Employee | 342 | | | 23.7% | 278 | 82.7% | ( 76.5% | – | 87.5% ) |
| | Self-employed | 1081 | | | 53.3% | 746 | 62.6% | ( 58.6% | – | 66.4% ) |
| | Non-working | 539 | | | 23.0% | 378 | 61.7% | ( 50.7% | – | 71.6% ) |
| Income | | | 924.61 | 3164.90 | | | | | | |
| | USD 0–120.8 | 502 | | | 25.2% | 325 | 58.5% | ( 51.3% | – | 65.3% ) |
| | USD 120.9–402.9 | 546 | | | 29.3% | 360 | 58.2% | ( 51.6% | – | 64.5% ) |
| | USD 120.9–402.9 | 412 | | | 22.7% | 319 | 78.2% | ( 71.7% | – | 83.5% ) |
| | USD 403.0–805.7 | 502 | | | 22.8% | 398 | 77.2% | ( 71.5% | – | 82.0% ) |
| Survey language | Dzongkha | 701 | | | 33.8% | 514 | 71.8% | ( 66.6% | – | 76.6% ) |
| | Tshanglakha | 680 | | | 30.8% | 494 | 70.7% | ( 63.3% | – | 77.2% ) |
| | Lhotshamkha | 547 | | | 33.2% | 368 | 58.6% | ( 50.8% | – | 66.1% ) |
| | English | 34 | | | 2.2% | 26 | 73.4% | ( 51.0% | – | 87.9% ) |
| Family history of hypertension | Negative | 1311 | | | 64.9% | 882 | 62.0% | ( 57.0% | – | 66.7% ) |
| | Positive | 651 | | | 35.1% | 520 | 76.7% | ( 71.9% | – | 80.9% ) |
| Tobacco use | Never use | 1556 | | | 74.3% | 1151 | 71.0% | ( 66.6% | – | 75.0% ) |
| | Currently use | 406 | | | 25.7% | 251 | 56.1% | ( 48.6% | – | 63.4% ) |
| Alcohol consumption | Never drink | 990 | | | 47.7% | 721 | 67.8% | ( 62.6% | – | 72.6% ) |
| | Light or moderate drinker | 548 | | | 30.3% | 379 | 66.7% | ( 59.1% | – | 73.6% ) |
| | Heavy drinker | 424 | | | 21.9% | 302 | 66.3% | ( 59.2% | – | 72.8% ) |
| Fruit and vegetables consumption | | | 4.52 | 3.48 | | | | | | |
| | Above 5 portions per day | 656 | | | 33.9% | 485 | 69.3% | ( 62.0% | – | 75.8% ) |
| | Below or 5 portions per day | 1306 | | | 66.1% | 917 | 66.0% | ( 61.3% | – | 70.5% ) |
| Physical activity | | | 475.97 | 682.04 | | | | | | |
| | Above or 150 mins per week | 1811 | | | 93.3% | 1295 | 67.0% | ( 62.5% | – | 71.2% ) |
| | Below 150 mins per week | 151 | | | 6.7% | 107 | 69.2% | ( 58.9% | – | 77.8% ) |
| Salt intake | | | 14.42 | 12.81 | | | | | | |
| | Below 5 g per day | 22 | | | 0.8% | 18 | 86.1% | ( 65.2% | – | 95.3% ) |
| | Above 5 g per day | 1940 | | | 99.2% | 1384 | 67.0% | ( 62.7% | – | 71.0% ) |
| Blood pressure[5] | Normal | 1068 | | | 58.0% | 705 | 62.4% | ( 57.0% | – | 67.5% ) |
| | High | 894 | | | 42.0% | 697 | 73.7% | ( 68.6% | – | 78.3% ) |

[1] Non-weighted calculation

[2] Complex sampling weight-adjusted %

[3] Participants who answered "Yes" to the questionnaire item "Have you ever had your blood pressure measured by a doctor or other health worker?"

[4] Estimated % adjusted with complex sampling: Participants with a previous blood pressure measurement/total target population

[5] "High" blood pressure was defined by the following criteria: (1) systolic blood pressure (SBP) ≥140 mmHg or diastolic blood pressure (DBP) ≥90 mmHg, measured as the mean of three measurements taken by health staff in the STEPS survey; (2) a previous diagnosis of hypertension by healthcare workers; (3) currently taking medication for hypertension.

to ensure they were representative of the national population [7] and adjusted based on the adjusted F-value, which is a variant of the second-order Rao–Scott adjusted chi-square statistic given the multistage random sampling [31, 32]. Variables with a $p$-value $<0.05$ were considered significant. In the study, participants aged 18–69 years were included. Those with missing data were excluded, and pregnant women were excluded because of health behavioral and biological differences.

In addition, given the gender differences and mother-child health programs, we also conducted a subgroup analysis for gender groups.

## Research ethics

The study protocol was approved by the Research Ethics Board of Health, Ministry of Health of the Royal Government of Bhutan (No. REBH/Approval/2018/089) and the Ethics Committee of the Kyoto University Hospital and Graduate School of Medicine (No. R1796). The above ethics committees waived the requirement for individual informed consent because this secondary analysis used only de-identified data. We have submitted a written pledge of confidentiality to the Ministry of Health of the Royal Government of Bhutan.

## Results

Our target participants included 1,962 individuals (Fig 1).

Table 1 shows the distribution of sociodemographic characteristics of all participants in this study. The target participants included 772 men (57.6%, standard error (SE): 1.9%, 95% CI: 53.7–61.4%) and 1,190 women (42.4%, SE: 1.9%, 95% CI: 38.6–46.3%). Most participants were married or cohabiting (83.1%, SE: 1.2%, 95% CI: 80.5–85.4%). The mean participant age was

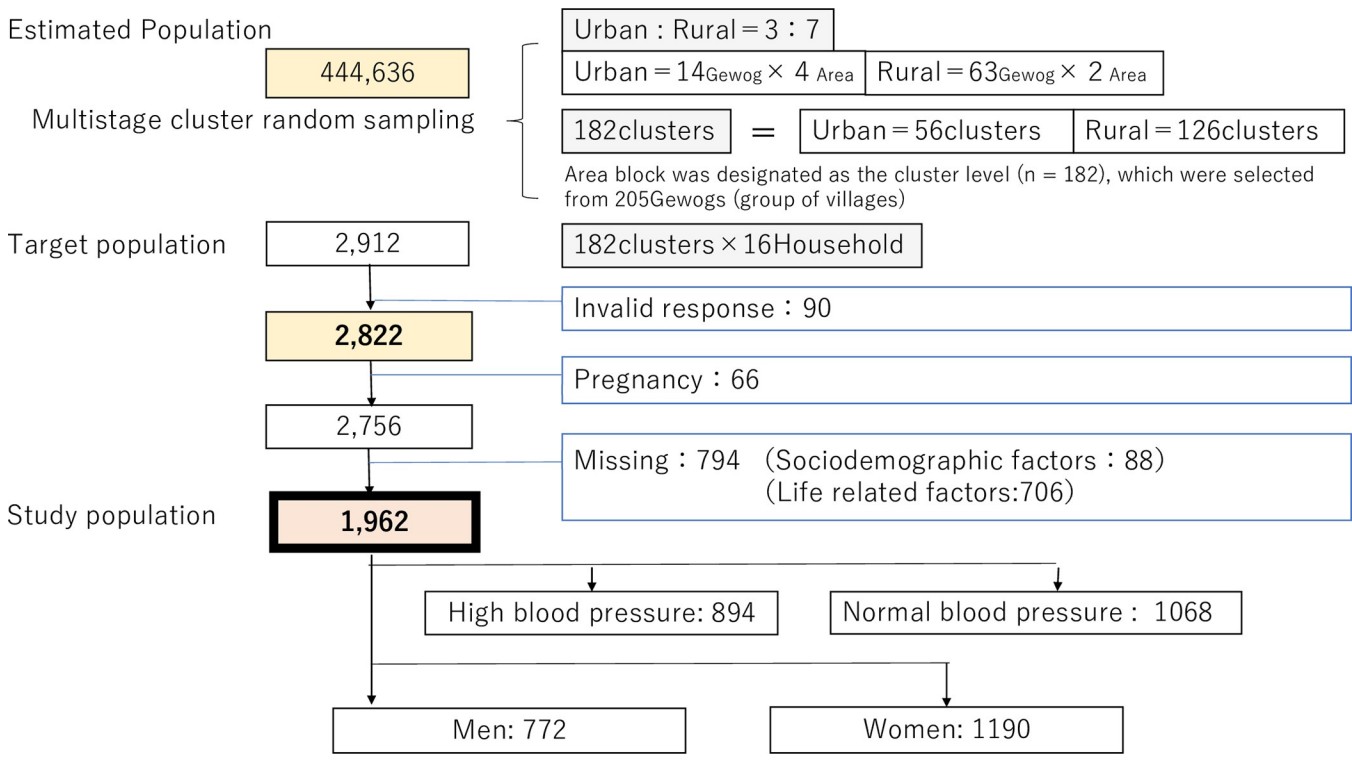

**Fig 1. Target population.**

40.25 years (non-weighted), 37.5 years (weighted) (SD: 12.4, SE: 0.407, 95% CI: 36.7–38.3). Over half of participants had no education (54.1%, SE: 2.1%, 95% CI: 50.0–58.2%) and were self-employed (53.3%, SE: 3.3%, 95% CI: 46.9–59.7%). Forty-two percent (SE: 1.7%, 95% CI: 38.7–45.4%) of all participants had hypertension, and 67.1% (SE: 2.1%, 95% CI: 62.9–71.2%) of participants had never had a blood pressure measurement. Among those with hypertension, 26.3% (crude n = 197, SE: 2.5%, 95% CI: 21.7–31.4%) had never had a blood pressure measurement before.

Table 2 shows the descriptive differences between those who had hypertension or not. In total, those who had hypertension had a higher coverage of blood pressure measurement. However, except for the category of salt intake (<5 g per day), no category had 100% coverage of blood pressure measurement before the NCD STEPS survey.

Among those who had normal blood pressure, current tobacco use, heavy alcohol drinker, less fruit and vegetables consumption, and high salt intake had less coverage of blood pressure measurement. However, a shortage of physical activity had a higher coverage of blood pressure measurement.

Table 3 shows the descriptive differences between men and women.

In total, women had an increased likelihood of previous blood pressure measurements than men.

Table 4 shows the univariable regression for blood pressure measurement for all participants. Women (odds ratio (OR): 2.019, 95% CI: 1.533–2.660) and those who were married or cohabiting (OR: 2.611, 95% CI: 1.731–3.937) or separated, divorced, or widowed (OR: 2.556, 95% CI: 1.439–4.540) had a significantly increased likelihood of previous blood pressure measurement. Those with an education level of 1–10 years (OR: 1.414, 95% CI: 1.104–1.811) also had an increased likelihood of previous blood pressure measurement compared with those with no formal education (OR: 1.000). Regarding working status, self-employed (OR: 0.351, 95% CI: 0.234–0.526) and non-working (OR: 0.338, 95% CI: 0.193–0.593) participants had a decreased likelihood of previous blood pressure measurement than employees (OR: 1.000). In terms of income, all participants in Quartile-4 (Nu. 60,001+: USD 805.8+ (OR: 2.404, 95% CI: 1.575–3.667)) and Quartile-3 (Nu. 30,001–60,000: USD 403.0–805.7 (OR: 2.546, 95% CI: 1.640–3.955)) had an increased likelihood of previous blood pressure measurement compared with those in the lowest income category, Quartile-1 (Nu. 0–9,000: USD 0–120.8 [OR: 1.000]). We found a significant association between Quartile-3, Quartile-4, and previous blood pressure measurement but not between Q2 (Nu. 9,001–30,000: USD 120.9–402.9) and previous blood pressure measurement.

In terms of gender differences, there were significant associations between previous blood pressure measurement and age, education, income, and working status (self-employed [OR: 0.219, 95% CI: 0.133–0.361], non-working [OR: 0.114, 95% CI: 0.050–0.263], employee [OR:1.000]) among men but not among women. There was a significant association between hypertension and income level (Quartile-2 [OR: 1.984, 95% CI: 1.209–3.255], Quartile-1 [OR: 2.161, 95% CI: 1.415–3.299], Quartile-4 [OR:1]) among women.

Those with a family history of hypertension (OR: 2.019, 95% CI: 1.549–2.633) and those with hypertension (OR: 1.695, 95% CI: 1.280–2.243) also had an increased likelihood of previous blood pressure measurement.

Table 5 shows the multivariable regression analysis for blood pressure measurement and behavioral factors for all participants. Tobacco use (adjusted odds ratio (AOR): 0.538, 95% CI: 0.380–0.761) and salt intake (AOR: 0.247, 95% CI: 0.068–0.893) were associated with a decreased likelihood of previous blood pressure measurement.

**Table 2. Differences between participants with hypertension or not and the distribution of socio-demographic characteristics and life behaviors for blood pressure measurement.**

| | | Hypertension[1] | | | | | | | Normal blood pressure | | | | | | |
|---|---|---|---|---|---|---|---|---|---|---|---|---|---|---|---|
| | | n | BP measured[2] | Estimated[3] | 95% CI | | | | n | BP measured[2] | Estimated[3] | 95% CI | | | |
| Gender | Men | 348 | 246 | 67.6% | ( | 59.9% | – | 74.5% | ) | 424 | 240 | 56.1% | ( | 48.3% | – | 63.6% | ) |
| | Women | 548 | 452 | 81.4% | ( | 77.0% | – | 85.1% | ) | 642 | 464 | 71.4% | ( | 66.3% | – | 76.0% | ) |
| Marital Status | Married or cohabiting | 735 | 576 | 74.5% | ( | 32.9% | – | 74.5% | ) | 115 | 62 | 43.8% | ( | 32.8% | – | 55.5% | ) |
| | Never married | 40 | 24 | 54.5% | ( | 69.1% | – | 79.2% | ) | 853 | 582 | 65.8% | ( | 59.6% | – | 71.5% | ) |
| | Separated, divorced, or widowed | 121 | 98 | 81.5% | ( | 70.5% | – | 89.0% | ) | 98 | 60 | 55.7% | ( | 42.0% | – | 68.7% | ) |
| Age | 18–29 years | 98 | 79 | 75.2% | ( | 62.2% | – | 84.9% | ) | 345 | 218 | 57.5% | ( | 50.4% | – | 64.3% | ) |
| | 30–39 years | 238 | 187 | 68.4% | ( | 59.0% | – | 76.5% | ) | 333 | 240 | 67.3% | ( | 59.3% | – | 74.4% | ) |
| | 40–49 years | 263 | 207 | 78.2% | ( | 71.3% | – | 83.7% | ) | 212 | 142 | 67.9% | ( | 59.9% | – | 75.0% | ) |
| | 50–59 years | 186 | 141 | 75.7% | ( | 65.4% | – | 83.8% | ) | 120 | 72 | 56.7% | ( | 46.5% | – | 66.3% | ) |
| | 60–69 years | 111 | 84 | 77.7% | ( | 66.7% | – | 85.8% | ) | 56 | 32 | 53.1% | ( | 37.4% | – | 68.3% | ) |
| Education | No formal education | 611 | 470 | 72.6% | ( | 65.9% | – | 78.5% | ) | 592 | 358 | 55.5% | ( | 48.5% | – | 62.2% | ) |
| | 1–10 years | 243 | 194 | 73.9% | ( | 65.1% | – | 81.1% | ) | 357 | 259 | 69.5% | ( | 62.7% | – | 75.6% | ) |
| | 11–12 years | 18 | 14 | 75.6% | ( | 43.3% | – | 92.6% | ) | 81 | 56 | 60.0% | ( | 44.2% | – | 74.0% | ) |
| | Above 12 years | 24 | 20 | 90.2% | ( | 73.5% | – | 96.8% | ) | 36 | 31 | 84.2% | ( | 63.2% | – | 94.3% | ) |
| Working status | Employee | 142 | 125 | 88.3% | ( | 81.0% | – | 93.0% | ) | 200 | 153 | 79.1% | ( | 70.5% | – | 85.7% | ) |
| | Self-employed | 524 | 392 | 68.4% | ( | 62.1% | – | 74.0% | ) | 557 | 354 | 57.9% | ( | 52.4% | – | 63.2% | ) |
| | Non-working | 230 | 181 | 73.1% | ( | 60.7% | – | 82.7% | ) | 309 | 197 | 54.4% | ( | 43.4% | – | 64.9% | ) |
| Income | USD 0–120.8 | 235 | 167 | 64.2% | ( | 55.5% | – | 72.1% | ) | 267 | 158 | 53.6% | ( | 45.0% | – | 62.1% | ) |
| | USD 120.9–402.9 | 262 | 200 | 68.6% | ( | 58.6% | – | 77.2% | ) | 284 | 160 | 50.5% | ( | 42.8% | – | 58.2% | ) |
| | USD 120.9–402.9 | 177 | 150 | 88.9% | ( | 81.5% | – | 93.5% | ) | 235 | 169 | 72.2% | ( | 62.4% | – | 80.3% | ) |
| | USD 403.0–805.7 | 222 | 181 | 78.5% | ( | 70.5% | – | 84.9% | ) | 280 | 217 | 76.1% | ( | 68.6% | – | 82.3% | ) |
| Survey language | Dzongkha | 279 | 224 | 78.0% | ( | 69.7% | – | 84.6% | ) | 422 | 290 | 67.5% | ( | 59.9% | – | 74.4% | ) |
| | Tshanglakha | 369 | 281 | 73.8% | ( | 65.0% | – | 81.0% | ) | 311 | 213 | 67.9% | ( | 58.1% | – | 76.3% | ) |
| | Lhotshamkha | 243 | 189 | 68.8% | ( | 58.7% | – | 77.4% | ) | 304 | 179 | 52.0% | ( | 43.7% | – | 60.1% | ) |
| | English | 5 | 4 | 87.5% | ( | 39.0% | – | 98.7% | ) | 29 | 22 | 71.3% | ( | 46.6% | – | 87.6% | ) |
| Family history of hypertension | Negative | 580 | 427 | 68.5% | ( | 61.5% | – | 74.7% | ) | 731 | 455 | 57.4% | ( | 51.5% | – | 63.0% | ) |
| | Positive | 316 | 271 | 83.1% | ( | 76.9% | – | 87.9% | ) | 335 | 249 | 71.9% | ( | 64.7% | – | 78.0% | ) |
| Tobacco use | Never use | 733 | 579 | 75.9% | ( | 70.6% | – | 80.5% | ) | 823 | 572 | 67.2% | ( | 61.7% | – | 72.3% | ) |
| | Currently use | 163 | 119 | 66.6% | ( | 55.8% | – | 75.9% | ) | 243 | 132 | 49.7% | ( | 40.5% | – | 59.0% | ) |
| Alcohol consumption | Never drink | 403 | 327 | 75.2% | ( | 68.8% | – | 80.7% | ) | 587 | 394 | 63.2% | ( | 56.7% | – | 69.2% | ) |
| | Light or moderate drinker | 258 | 190 | 69.9% | ( | 60.3% | – | 78.0% | ) | 290 | 189 | 64.3% | ( | 54.5% | – | 73.1% | ) |
| | Heavy drinker | 235 | 181 | 76.0% | ( | 68.0% | – | 82.4% | ) | 189 | 121 | 57.3% | ( | 46.8% | – | 67.3% | ) |
| Fruit and vegetables consumption | Above 5 portions per day | 319 | 245 | 71.2% | ( | 61.9% | – | 79.0% | ) | 337 | 240 | 67.8% | ( | 58.2% | – | 76.1% | ) |
| | Below or 5 portions per day | 577 | 453 | 75.1% | ( | 69.5% | – | 80.0% | ) | 729 | 464 | 59.7% | ( | 53.9% | – | 65.2% | ) |
| Physical activity | Above or 150 mins per week | 832 | 649 | 73.8% | ( | 68.3% | – | 78.7% | ) | 979 | 646 | 62.0% | ( | 56.5% | – | 67.3% | ) |
| | Below 150 mins per week | 64 | 49 | 72.5% | ( | 57.4% | – | 83.7% | ) | 87 | 58 | 67.0% | ( | 53.3% | – | 78.3% | ) |

*(Continued)*

**Table 2.** (Continued)

| | | Hypertension[1] | | | | | | | Normal blood pressure | | | | | | |
| --- | --- | --- | --- | --- | --- | --- | --- | --- | --- | --- | --- | --- | --- | --- | --- |
| | | n | BP measured[2] | Estimated[3] | 95% CI | | | | | n | BP measured[2] | Estimated[3] | 95% CI | | | |
| Salt intake | Below 5 g per day | 10 | 10 | 100.0% | ( | 100.0% | – | 100.0% | ) | 12 | 8 | 76.5% | ( | 46.1% | – | 92.5% | ) |
| | Above or 5 g per day | 886 | 688 | 73.5% | ( | 68.3% | – | 78.1% | ) | 1054 | 696 | 62.2% | ( | 56.8% | – | 67.4% | ) |

[1] Hypertension was defined by the following criteria: (1) systolic blood pressure (SBP) ≥140 mmHg or diastolic blood pressure (DBP) ≥90 mmHg, measured as the mean of three measurements taken by health staff in the STEPS survey; (2) a previous diagnosis of hypertension by healthcare workers; (3) currently taking medication for hypertension.

[2] Participants who answered "Yes" to the questionnaire item "Have you ever had your blood pressure measured by a doctor or other health worker?"

[3] Estimated % adjusted with complex sampling: Participants with a previous blood pressure measurement/total target population

## Discussion

### Inadequate blood pressure screening

Participants with normal blood pressure and 26.3% of those who had hypertension had never undergone blood pressure measurement before the STEPS survey. This may suggest that screening for blood pressure was inadequate. Blood pressure measurement is necessary in people with hypertension or people at high risk of hypertension. Early detection and treatment of hypertension can be effective in preventing other diseases [3, 17]. Some of the risk factors for hypertension are preventable, whereas some, such as aging or genetics, are not [9, 17].

A previous qualitative study in Bhutan indicated that there were people who had never undergone blood pressure measurement although they had high blood pressure; however, this previous qualitative study could not show that these results were representative of Bhutan more widely due to the small sample size from a small area [10]. Using a national survey with sample sizes representative of Bhutan [7], we could demonstrate the importance of early detection of hypertension. People with hypertension remain undiagnosed for multiple reasons, including behavioral reasons, lack of awareness, limited access to the health system, patient projections, or medical costs [33–35]. Medical tests should be conducted given their cost-effectiveness. If all the people with no previous blood pressure measurement had normal blood pressure, there would not be a problem. However, this is not the case, so a selective approach is needed to identify people with hypertension or people at high risk for hypertension and which simultaneously does not waste scarce resources.

### Social background factors and gender differences

Married women had more experience with blood pressure measurement than men. For men, aging and working environment were significantly associated with previous blood pressure measurement. However, for women, except for income, there was no significant association between previous blood pressure measurement and working status or aging (Table 4). The socioeconomic factors associated with blood pressure measurement may differ between men and women.

In this study, we could not determine the cause of the gender differences observed in blood pressure measurement; however, existing maternal and child health programs might be the reason that women with a married or cohabiting status had an increased prevalence of previous blood pressure measurement [36, 37]. In Bhutan, maternal and child health services have improved markedly in the past two decades [37, 38]. The government of Bhutan provides free

**Table 3. Gender differences in the distribution of the socio-demographic characteristics and life behaviors for blood pressure measurement.**

| | | Men | | | | | | | Women | | | | | | |
|---|---|---|---|---|---|---|---|---|---|---|---|---|---|---|---|
| | | n | BP measured[1] | Estimated[2] | 95% CI | | | | n | BP measured[1] | Estimated[2] | 95% CI | | | |
| Marital status | Married or cohabiting | 652 | 423 | 63.2% | ( | 56.5% | – | 69.4% | ) | 936 | 735 | 78.6% | ( | 74.9% | – | 81.8% | ) |
| | Never married | 80 | 41 | 45.0% | ( | 33.2% | – | 57.3% | ) | 75 | 45 | 50.4% | ( | 35.8% | – | 64.8% | ) |
| | Separated, divorced, or widowed | 40 | 22 | 58.5% | ( | 38.8% | – | 75.8% | ) | 179 | 136 | 73.3% | ( | 63.8% | – | 81.1% | ) |
| Age | 18–29 years | 142 | 69 | 48.9% | ( | 38.1% | – | 59.7% | ) | 301 | 228 | 74.6% | ( | 68.3% | – | 80.0% | ) |
| | 30–39 years | 223 | 152 | 63.1% | ( | 53.9% | – | 71.4% | ) | 348 | 275 | 75.8% | ( | 70.1% | – | 80.7% | ) |
| | 40–49 years | 192 | 130 | 71.0% | ( | 63.4% | – | 77.6% | ) | 283 | 219 | 76.3% | ( | 68.7% | – | 82.4% | ) |
| | 50–59 years | 137 | 84 | 61.3% | ( | 52.1% | – | 69.8% | ) | 169 | 129 | 78.1% | ( | 68.4% | – | 85.5% | ) |
| | 60–69 years | 78 | 51 | 66.0% | ( | 53.3% | – | 76.7% | ) | 89 | 65 | 76.2% | ( | 62.7% | – | 85.9% | ) |
| Education | No formal education | 383 | 214 | 52.1% | ( | 43.7% | – | 60.3% | ) | 820 | 614 | 74.4% | ( | 70.3% | – | 78.0% | ) |
| | 1–10 years | 297 | 205 | 67.9% | ( | 60.5% | – | 74.6% | ) | 303 | 248 | 77.9% | ( | 71.1% | – | 83.5% | ) |
| | 11–12 years | 50 | 31 | 55.3% | ( | 38.4% | – | 71.1% | ) | 49 | 39 | 80.0% | ( | 63.4% | – | 90.3% | ) |
| | Above 12 years | 42 | 36 | 87.2% | ( | 69.6% | – | 95.3% | ) | 1 | 15 | 84.0% | ( | 58.6% | – | 95.1% | ) |
| Working status | Employee | 246 | 198 | 83.2% | ( | 76.0% | – | 88.6% | ) | 96 | 80 | 79.9% | ( | 66.2% | – | 89.0% | ) |
| | Self-employed | 424 | 239 | 52.1% | ( | 46.2% | – | 57.9% | ) | 657 | 507 | 76.5% | ( | 72.6% | – | 80.0% | ) |
| | Non-working | 102 | 49 | 36.1% | ( | 22.5% | – | 52.5% | ) | 437 | 329 | 73.7% | ( | 66.9% | – | 79.6% | ) |
| Income | USD 0–120.8 | 179 | 94 | 48.7% | ( | 38.5% | – | 59.0% | ) | 323 | 231 | 70.0% | ( | 64.1% | – | 75.2% | ) |
| | USD 120.9–402.9 | 219 | 123 | 49.2% | ( | 39.4% | – | 59.0% | ) | 327 | 237 | 70.6% | ( | 64.1% | – | 76.4% | ) |
| | USD 120.9–402.9 | 167 | 119 | 75.5% | ( | 66.6% | – | 82.7% | ) | 245 | 200 | 82.2% | ( | 75.1% | – | 87.6% | ) |
| | USD 403.0–805.7 | 207 | 150 | 72.8% | ( | 63.6% | – | 80.3% | ) | 295 | 248 | 83.4% | ( | 78.2% | – | 87.6% | ) |
| Survey language | Dzongkha | 259 | 184 | 70.2% | ( | 62.4% | – | 76.9% | ) | 442 | 330 | 73.9% | ( | 68.7% | – | 78.6% | ) |
| | Tshanglakha | 238 | 152 | 65.0% | ( | 52.9% | – | 75.4% | ) | 442 | 342 | 77.2% | ( | 71.5% | – | 82.1% | ) |
| | Lhotshamkha | 255 | 135 | 48.2% | ( | 38.8% | – | 57.8% | ) | 292 | 233 | 76.1% | ( | 68.2% | – | 82.5% | ) |
| | English | 20 | 15 | 70.6% | ( | 42.0% | – | 88.8% | ) | 14 | 11 | 83.3% | ( | 46.5% | – | 96.6% | ) |
| Family history of hypertension | Negative | 528 | 307 | 55.0% | ( | 47.9% | – | 61.9% | ) | 783 | 575 | 71.8% | ( | 67.2% | – | 76.1% | ) |
| | Positive | 244 | 179 | 72.0% | ( | 64.0% | – | 78.8% | ) | 407 | 341 | 82.7% | ( | 77.2% | – | 87.1% | ) |
| Tobacco use | Never use | 525 | 347 | 64.4% | ( | 57.1% | – | 71.0% | ) | 1031 | 804 | 77.9% | ( | 74.2% | – | 81.1% | ) |
| | Currently use | 247 | 139 | 53.8% | ( | 45.0% | – | 62.3% | ) | 159 | 112 | 63.6% | ( | 53.5% | – | 72.6% | ) |
| Alcohol consumption | Never drink | 287 | 183 | 58.9% | ( | 50.0% | – | 67.2% | ) | 703 | 538 | 75.2% | ( | 70.8% | – | 79.2% | ) |
| | Light or moderate drinker | 278 | 172 | 63.9% | ( | 54.0% | – | 72.9% | ) | 270 | 207 | 73.2% | ( | 65.9% | – | 79.4% | ) |
| | Heavy drinker | 207 | 131 | 59.1% | ( | 50.3% | – | 67.3% | ) | 217 | 171 | 81.0% | ( | 73.1% | – | 87.0% | ) |
| Fruit and vegetables consumption | Above 5 portions per day | 270 | 170 | 63.3% | ( | 51.8% | – | 73.4% | ) | 386 | 315 | 79.1% | ( | 72.4% | – | 84.6% | ) |
| | Below or 5 portions per day | 502 | 316 | 59.4% | ( | 52.8% | – | 65.6% | ) | 804 | 601 | 74.3% | ( | 70.1% | – | 78.2% | ) |
| Physical activity | Above or 150 mins per week | 733 | 460 | 60.7% | ( | 54.1% | – | 66.9% | ) | 1078 | 835 | 76.2% | ( | 72.6% | – | 79.4% | ) |
| | Below 150 mins per week | 39 | 26 | 62.4% | ( | 43.0% | – | 78.4% | ) | 112 | 81 | 72.7% | ( | 61.6% | – | 81.5% | ) |
| Salt intake | Below 5 g per day | 10 | 8 | 81.3% | ( | 45.0% | – | 95.9% | ) | 12 | 10 | 90.5% | ( | 65.5% | – | 97.9% | ) |
| | Above or 5 g per day | 762 | 478 | 60.6% | ( | 54.3% | – | 66.6% | ) | 1178 | 906 | 75.6% | ( | 72.2% | – | 78.8% | ) |
| Blood pressure[3] | Normal | 424 | 240 | 56.1% | ( | 48.3% | – | 63.6% | ) | 642 | 464 | 71.4% | ( | 66.3% | – | 76.0% | ) |
| | High | 348 | 246 | 67.6% | ( | 59.9% | – | 74.5% | ) | 548 | 452 | 81.4% | ( | 77.0% | – | 85.1% | ) |

[1] Participants who answered "Yes" to the questionnaire item "Have you ever had your blood pressure measured by a doctor or other health worker?"

[2] Estimated % adjusted with complex sampling: Participants with a previous blood pressure measurement/total target population

**Table 4. Univariable logistic regression of previous blood pressure measurement in the target population.**

| | | Total | | | | | | | | Men | | | | | | | | Women | | | | | | |
|---|---|---|---|---|---|---|---|---|---|---|---|---|---|---|---|---|---|---|---|---|---|---|---|---|---|
| | | OR | \multicolumn 95% CI | | | | | P-value | OR | 95% CI | | | | | | P-value | OR | 95% CI | | | | | | P-value |
| Gender | Men | Ref | ( | | – | | ) | | **0.000** | Ref | ( | | – | – | – | ) | – | Ref | ( | – | – | – | ) | – |
| | Women | **2.019** | ( | 1.533 | – | 2.660 | ) | | | – | ( | – | – | – | ) | – | – | ( | – | – | – | ) | – |
| Marital Status | Never married | Ref | ( | | – | | ) | | **0.000** | Ref | ( | | – | | ) | **0.010** | Ref | ( | | – | | ) | **0.000** |
| | Married or cohabiting | **2.611** | ( | 1.731 | – | 3.937 | ) | | | **2.101** | ( | 1.253 | – | 3.524 | ) | | **3.611** | ( | 1.906 | – | 6.842 | ) | |
| | Separated, divorced, or widowed | **2.556** | ( | 1.439 | – | 4.540 | ) | | | **1.724** | ( | 0.669 | – | 4.441 | ) | | **2.708** | ( | 1.244 | – | 5.896 | ) | |
| Age | 18–29 years | Ref | ( | | – | | ) | | **0.038** | Ref | ( | | – | | ) | **0.004** | Ref | ( | | – | | ) | 0.968 |
| | 30–39 years | 1.318 | ( | 0.971 | – | 1.789 | ) | | | **1.790** | ( | 1.114 | – | 2.875 | ) | | 1.068 | ( | 0.726 | – | 1.57 | ) | |
| | 40–49 years | **1.726** | ( | 1.249 | – | 2.385 | ) | | | **2.559** | ( | 1.507 | – | 4.348 | ) | | 1.094 | ( | 0.664 | – | 1.803 | ) | |
| | 50–59 years | 1.371 | ( | 0.909 | – | 2.069 | ) | | | 1.658 | ( | 0.958 | – | 2.871 | ) | | 1.216 | ( | 0.683 | – | 2.168 | ) | |
| | 60–69 years | 1.466 | ( | 0.905 | – | 2.375 | ) | | | **2.032** | ( | 1.058 | – | 3.900 | ) | | 1.089 | ( | 0.548 | – | 2.164 | ) | |
| Education | No formal education | Ref | ( | | – | | ) | | **0.008** | Ref | ( | | – | | ) | **0.000** | Ref | ( | | – | | ) | 0.575 |
| | 1–10 years | **1.414** | ( | 1.104 | – | 1.811 | ) | | | **1.951** | ( | 1.376 | – | 2.765 | ) | | 1.218 | ( | 0.812 | – | 1.826 | ) | |
| | 11–12 years | 0.975 | ( | 0.534 | – | 1.778 | ) | | | 1.140 | ( | 0.539 | – | 2.410 | ) | | 1.382 | ( | 0.577 | – | 3.314 | ) | |
| | Above12 years | **3.677** | ( | 1.483 | – | 9.119 | ) | | | **6.282** | ( | 2.029 | – | 19.452 | ) | | 1.805 | ( | 0.494 | – | 6.603 | ) | |
| Working status | Employee | Ref | ( | | – | | ) | | **0.000** | Ref | ( | | – | | ) | **0.000** | Ref | ( | | – | | ) | 0.587 |
| | Self-employed | **0.351** | ( | 0.234 | – | 0.526 | ) | | | **0.219** | ( | 0.133 | – | 0.361 | ) | | 0.820 | ( | 0.406 | – | 1.656 | ) | |
| | Non-working | **0.338** | ( | 0.193 | – | 0.593 | ) | | | **0.114** | ( | 0.050 | – | 0.263 | ) | | 0.707 | ( | 0.324 | – | 1.545 | ) | |
| Income | USD 0–120.8 | Ref | ( | | – | | ) | | **0.000** | Ref | ( | | – | | ) | **0.000** | Ref | ( | | – | | ) | **0.001** |
| | USD 120.9–402.9 | 0.989 | ( | 0.714 | – | 1.372 | ) | | | 1.021 | ( | 0.606 | – | 1.718 | ) | | 1.033 | ( | 0.710 | – | 1.502 | ) | |
| | USD 120.9–402.9 | **2.546** | ( | 1.640 | – | 3.955 | ) | | | **3.250** | ( | 1.768 | – | 5.974 | ) | | **1.984** | ( | 1.209 | – | 3.255 | ) | |
| | USD 403.0–805.7 | **2.404** | ( | 1.575 | – | 3.667 | ) | | | **2.816** | ( | 1.524 | – | 5.203 | ) | | **2.161** | ( | 1.415 | – | 3.299 | ) | |
| Tobacco use | Never use | Ref | ( | | – | | ) | | **0.000** | Ref | ( | | – | | ) | **0.035** | Ref | ( | | – | | ) | **0.003** |
| | Currently use | **0.523** | ( | 0.378 | – | 0.724 | ) | | | **0.643** | ( | 0.427 | – | 0.969 | ) | | **0.496** | ( | 0.314 | – | 0.785 | ) | |
| Alcohol consumption | Never drink | Ref | ( | | – | | ) | | 0.925 | Ref | ( | | – | | ) | 0.586 | Ref | ( | | – | | ) | 0.280 |
| | Light or moderate drinker | 0.953 | ( | 0.662 | – | 1.372 | ) | | | 1.239 | ( | 0.754 | – | 2.036 | ) | | 0.897 | ( | 0.600 | – | 1.342 | ) | |
| | Heavy drinker | 0.936 | ( | 0.663 | – | 1.322 | ) | | | 1.008 | ( | 0.652 | – | 1.556 | ) | | 1.405 | ( | 0.849 | – | 2.326 | ) | |
| Fruit and vegetables consumption | Above 5 portions per day | Ref | ( | | – | | ) | | 0.408 | Ref | ( | | – | | ) | 0.525 | Ref | ( | | – | | ) | 0.222 |
| | Below or 5 portions per day | 0.862 | ( | 0.605 | – | 1.228 | ) | | | 0.848 | ( | 0.509 | – | 1.413 | ) | | 0.764 | ( | 0.496 | – | 1.178 | ) | |
| Physical activity | Above or 150 mins per week | Ref | ( | | – | | ) | | 0.681 | Ref | ( | | – | | ) | 0.871 | Ref | ( | | – | | ) | 0.488 |
| | Below 150 mins per week | 1.104 | ( | 0.687 | – | 1.775 | ) | | | 1.072 | ( | 0.461 | – | 2.492 | ) | | 0.832 | ( | 0.493 | – | 1.403 | ) | |
| Salt intake | Below 5 g per day | Ref | ( | | – | | ) | | 0.066 | Ref | ( | | – | | ) | 0.218 | Ref | ( | | – | | ) | 0.175 |
| | Above or 5 g per day | 0.328 | ( | 0.100 | – | 1.079 | ) | | | 0.354 | ( | 0.067 | – | 1.859 | ) | | 0.327 | ( | 0.065 | – | 1.651 | ) | |
| Family history of hypertension | Negative | Ref | ( | | – | | ) | | **0.000** | Ref | ( | | – | | ) | **0.000** | Ref | ( | | – | | ) | **0.004** |
| | Positive | **2.019** | ( | 1.549 | – | 2.633 | ) | | | **2.107** | ( | 1.427 | – | 3.112 | ) | | **1.87** | ( | 1.227 | – | 2.849 | ) | |
| High blood pressure[1] | Normal | Ref | ( | | – | | ) | | **0.000** | Ref | ( | | – | | ) | **0.011** | Ref | ( | | – | | ) | N/A |
| | High | **1.695** | ( | 1.280 | – | 2.243 | ) | | | **1.639** | ( | 1.122 | – | 2.394 | ) | | | ( | | – | | ) | |

[1] "High" blood pressure was defined by the following criteria: (1) systolic blood pressure (SBP) ≥140 mmHg or diastolic blood pressure (DBP) ≥90 mmHg, measured as the mean of three measurements taken by health staff in the STEPS survey; (2) a previous diagnosis of hypertension by healthcare workers; (3) currently taking medication for hypertension.

N/A: Not able to calculate systematically

**Table 5. Multivariable logistic regression analysis of previous blood pressure measurement and life-related behavioral factors.**

| Explanatory variables | | | AOR | 95% CI | | | | | | P-value[1)] |
|---|---|---|---|---|---|---|---|---|---|---|
| Model-1 | Tobacco use | Never use | | ( | | – | | | ) | 0.001 |
| | | Currently use | **0.538** | ( | 0.380 | – | 0.761 | | ) | |
| Model-2 | Alcohol consumption | Never drink | | ( | | – | | | ) | 0.994 |
| | | Light or moderate drinker | 1.010 | ( | 0.728 | – | 1.400 | | ) | |
| | | Heavy drinker | 0.989 | ( | 0.713 | – | 1.371 | | ) | |
| Model-3 | Fruit and vegetables consumption | Above 5 portions per day | | ( | | – | | | ) | 0.676 |
| | | Below or 5 portions per day | 0.934 | ( | 0.678 | – | 1.287 | | ) | |
| Model-4 | Physical activity | Above or 150 mins per week | | ( | | – | | | ) | 0.625 |
| | | Below 150 mins per week | 0.923 | ( | 0.670 | – | 1.273 | | ) | |
| Model-5 | Salt intake | Below 5 g per day | | ( | | – | | | ) | **0.033** |
| | | Above or 5 g per day | **0.247** | ( | 0.068 | – | 0.893 | | ) | |

Model-1 was adjusted for Age, Education, Family history of hypertension, Gender, Income, Marital status, Working status

Model-2 was adjusted for Education, Family history of hypertension, Gender, Income, Survey language, Working status

Model-3 was adjusted for Education, Family history of hypertension, Income, Marital status, Survey language, Working status

Model-4 was adjusted for Age, Education, Income, Marital status, Survey language, Working status

Model-5 was adjusted for Education, Family history of hypertension, Income, Marital status, Survey language, Working status

[1)] *P*-value: Bonferroni

access to basic public health services for all [12]. Nonetheless, among women, those in the above-average income category had a higher coverage of previous blood pressure measurement than those in the below-average income category. This difference might be explained by the travel cost or time cost involved in going to blood pressure measurement appointments [11, 39]. The number of PHCs has increased, and access to health care has improved over the past 30 years [37–40]. An additional 87.7% of the population had access to a PHC within two hours in 2019 [41]. However, traveling for two hours on a mountain road in Bhutan may be difficult for women. Lower-income women might also experience some barriers to visiting the hospital, as previous studies in other countries have indicated, such as insufficient education, gender gap problems, and insufficient family support [42].

We should pay more attention to the low-income group because we need to reduce barriers to accessing health care and promote blood pressure measurement among this group. Previous studies in other countries have also indicated that health-seeking behavior differs by gender [42, 43]. Women have greater awareness and higher consciousness of the treatments for hypertension but simultaneously experienced more barriers to visiting the hospital than men [42–45].

In Bhutan, there are fewer paternal programs for men than there are mother and child health programs; therefore, men might have less chance of having their health screened while healthy. For men, aging is a risk factor for hypertension [46]; therefore, it may be that men tend to go to the hospital after facing health problems as they age. If so, including men in the existing maternal and child health care programs could increase their opportunities to have their blood pressure monitored, encouraging them to pay attention to their own health and the health of their wives, children, and families. Working status had a significant association with the experience of blood pressure measurement among men. Those who are employed may have more opportunities to have their health examined and greater access to other services from their companies than those who are self-employed or non-working [47]. It is possible that housemen or self-employed men must work continuously to obtain a daily income, making it impossible to leave work and seek medical attention, or they may not be able to seek

medical attention without other initiatives. Those who are employed may have received some health care through their employer or colleagues. While workplace-based interventions can be effective [45–49], especially among men, methods of providing preventive health management to self-employed, non-working, and low-income people is an issue that remains to be addressed.

## Health behavior and risk perception

People with a family history of hypertension or those with hypertension had more experiences with blood pressure measurement in all analyses most likely because they have an increased awareness of the health threats they might face. People with an unhealthy lifestyle may not be aware of health threats or perceived health vulnerability, and one of the characteristics of people with a healthy lifestyle may be the ability to perceive health threats, regardless of whether they have pre-existing health problems or not. People with an unhealthy lifestyle (high salt intake or tobacco use) had less experience with blood pressure measurement.

Another previous study also showed that risk perception and preventive behavior are closely related [49]. Health perception of hypertension might also be related to health behavior [27, 50].

From the point of view of prevention, it is necessary to promote more blood pressure measurements to those who are at high risk or who have modifiable risk factors. In rural Bhutan, health education and health-related meetings often involve household representatives gathering in PHCs to provide knowledge. This is an effective means of communication due to the characteristics of strong family cohesion in Bhutan [10]. Furthermore, when developing and improving prevention programs, it may be necessary not only to make healthy people healthier but also to approach those at high risk of lifestyle-related diseases effectively and selectively. This selection should occur regardless of whether people are health conscious or not, because 26.3% of subjects in the STEPS survey had never had their blood pressure monitored before.

## Strengths and limitations of this study

This study was rare in that it focused on factors that are related to the experience of blood pressure measurement in Bhutan after adjusting for all possible variables and available confounders. Our study had several strengths. First, the dataset of the "National Survey for Noncommunicable Disease Risk Factors and Mental Health using WHO STEPS Approach in Bhutan– 2014" was representative of the whole of Bhutan, with a 97% valid response rate and the adoption of cluster random sampling methods. Second, the questionnaire included some biomarkers to adjust for risk factors for high blood pressure. Regarding limitations, first, a wish by participants to provide socially desirable answers could have affected the results due to face-to-face interviews. Nonetheless, this was the only method available to collect data from a low literacy rate target population. Second, we cannot deny that there were unmeasurable confounders for some associations that we found in this study, although we tried to adjust for all possible confounders from previous studies.

## Conclusions

To contribute to the early detection of hypertension, we investigated the rate of blood pressure measurement and associated factors in Bhutan. Participants with unhealthy lifestyles had less experience with blood pressure measurement. There was a gender gap in which social determinant factors were associated with previous blood pressure measurement. It is necessary to promote the early detection of hypertension and to pay more attention to low-income women, the self-employed, and non-working men. Levels of blood pressure screening were generally

inadequate. Individuals with hypertension or those at high risk for hypertension, such as one-quarter of subjects who had never had their blood pressure monitored before and those who lacked health consciousness, should not be overlooked.

## Supporting information

**S1 File. Sample size, power calculation, weighting, and adjusting the cluster random sampling procedure.**
(PDF)

**S2 File. Directed acyclic graph for multivariable logistic regression.**
(PDF)

## Acknowledgments

We would like to thank Cambridge English Correction Service for English language editing.

We appreciate the Khesar Gyalpo University of Medical Sciences of Bhutan, the Ministry of Health, Bhutan, and all the support from the Graduate School of Medicine, Kyoto University, Japan.

We used dagitty (http://www.dagitty.net/dags.html 3.0: released 2019-01-09) to evaluate a minimal sufficient adjustment set for total effect, models 1–5 (Table 5).

## Author Contributions

**Conceptualization:** Hiromi Kohori Segawa, Hironori Uematsu, Nidup Dorji, Ugyen Wangdi, Ryota Sakamoto, Yuichi Imanaka.

**Data curation:** Pemba Yangchen.

**Formal analysis:** Hiromi Kohori Segawa.

**Funding acquisition:** Hiromi Kohori Segawa, Yuichi Imanaka.

**Investigation:** Hiromi Kohori Segawa.

**Methodology:** Hiromi Kohori Segawa, Hironori Uematsu, Susumu Kunisawa.

**Project administration:** Hiromi Kohori Segawa, Nidup Dorji, Ugyen Wangdi, Chencho Dorjee, Pemba Yangchen.

**Supervision:** Chencho Dorjee, Yuichi Imanaka.

**Visualization:** Hiromi Kohori Segawa.

**Writing – original draft:** Hiromi Kohori Segawa.

**Writing – review & editing:** Hironori Uematsu, Nidup Dorji, Ugyen Wangdi, Chencho Dorjee, Pemba Yangchen, Susumu Kunisawa, Ryota Sakamoto, Yuichi Imanaka.

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
