## [Decision Letter · Decision Letter 0]

21 Jul 2021

PONE-D-21-12027

Factors related to blood pressure measurement, with attention needed for unemployed men, poor women, and people’s health behaviors: A cross-sectional study in Bhutan

PLOS ONE

Dear Dr. Imanaka,

Thank you for submitting your manuscript to PLOS ONE. After careful consideration, we feel that it has merit but does not fully meet PLOS ONE’s publication criteria as it currently stands. Therefore, we invite you to submit a revised version of the manuscript that addresses the points raised during the review process.

ACADEMIC EDITOR: 

The manuscript is interesting but will require further reworking and a major revision.<o:p></o:p>

While they recognize the potential interest of the subject studied, the reviewers raised a number of important issues that need to be properly addressed.<o:p></o:p>

We look forward to receiving your revised manuscript.

Kind regards,

Marcelo Arruda Nakazone, M.D., Ph.D.

Academic Editor

PLOS ONE

2. Thank you for including your ethics statement:  "The Research Ethics Boards of Health, Ministry of Health of the Royal Government of Bhutan (No. REBH/Approval/2018/089), and the Kyoto University Hospital and Graduate School of Medicine, Ethics Committee (No. R1796) approved the study protocol.".  

Please provide additional details regarding participant consent. In the ethics statement in the Methods and online submission information, please ensure that you have specified (1) whether consent was informed and (2) what type you obtained (for instance, written or verbal, and if verbal, how it was documented and witnessed). If your study included minors, state whether you obtained consent from parents or guardians. If the need for consent was waived by the ethics committee, please include this information.

3. In your Methods section, please provide additional information about the participant recruitment method and the demographic details of your participants. Please ensure you have provided sufficient details to replicate the analyses such as:

- the recruitment date range (month and year)

- a description of any inclusion/exclusion criteria that were applied to participant recruitment

- a statement as to whether your sample can be considered representative of a larger population

- a description of how participants were recruited

- descriptions of where participants were recruited and where the research took place."

4. Please provide a sample size and power calculation in the Methods, or discuss the reasons for not performing one before study initiation.

6. Thank you for stating the following in your Competing Interests section: 

“NO”

7. We note that you have indicated that data from this study are available upon request. PLOS only allows data to be available upon request if there are legal or ethical restrictions on sharing data publicly. For more information on unacceptable data access restrictions, please see http://journals.plos.org/plosone/s/data-availability#loc-unacceptable-data-access-restrictions.

Additional Editor Comments (if provided):

Reviewers' comments:

Reviewer's Responses to Questions

**Comments to the Author**

1. Is the manuscript technically sound, and do the data support the conclusions?

Reviewer #1: No

Reviewer #2: Yes

Reviewer #3: Partly

2. Has the statistical analysis been performed appropriately and rigorously? 

Reviewer #1: No

Reviewer #2: Yes

Reviewer #3: I Don't Know

3. Have the authors made all data underlying the findings in their manuscript fully available?

Reviewer #1: No

Reviewer #2: No

Reviewer #3: Yes

4. Is the manuscript presented in an intelligible fashion and written in standard English?

Reviewer #1: No

Reviewer #2: Yes

Reviewer #3: Yes

5. Review Comments to the Author

Reviewer #1: The major problems is the methodology and analysis done did not fit the study objectives. The study objective is to determine the factors associated with blood pressure measurement. It is not clear in the method the operational definition of blood pressure measurement in this study. What is the main outcome in this study? Is it behaviour practice of BP measurement for the past? duration or ever checked BP?

Reviewer #2: Reviewer Comments

Kohori-Segawa et al conducted a study to determine what characteristics are associated in Bhutanese people with the experience of having a blood pressure measurement. I find the article very interesting and neatly conducting. I have a few questions regarding the design:

1) In line x you mention patients with hypertension that had not had blood pressure measured, can you specify then how was the diagnosis made? Do you have any information regarding this? You mention this again in the conclusions in which you say people with hypertension have never had blood pressure measurements in their life so I do not understand how they know their hypertension status. Is it possible that they do not remember having their blood pressure measured rather than the measurement not being performed?

2) Regarding multivariable analysis, I would like for you to specify adjustment variables rather than just listing the variable category (line 162)

3) In line 317 you mention than unemployed or self employed men are less likely to have their blood pressure measured. A possible explanation could be the need to work on a daily basis to obtain income which makes it impossible to leave and seek for medical attention. I would like to know your thoughts about this.

Finally I would suggest changing the wording in sentences mentioning previous studies.

Reviewer #3: 1-Line 184: Can't find Figure1?. Does it refer to Table1?

2- It is difficult to understand the % in last column in table 1. How these % were calculated?

3-How to identify potential confounders? What are the potential confounders' variables to be included in multivariate analysis?

4-I am not English native speaker, but I just feel that there might be a need to further improvement in English writing fashion?

5-Title: It is too long, it should be revised?

6-Income: Readers who are not from the Kingdom of Bhutan may not understand like "Nu.0–9,000; ...." Is it possible to convert into a commonly known like UDS?

7-Discussion: Authors raised about "Considering barriers......" (line 284-285) and "Contradictions in health behavior...." (line 320), these are all the factors associated with blood pressure measure. The words .... barriers, contradictions might not appropriate to be used here?

6. PLOS authors have the option to publish the peer review history of their article (what does this mean?). If published, this will include your full peer review and any attached files.

Reviewer #1: No

Reviewer #2: **Yes: **Mercedes Aguilar-Soto

Reviewer #3: No

---

## [Author Response · Author response to Decision Letter 0]

3 Sep 2021

Dear Editor and reviewers,

We really appreciate your review and feedback during these difficult times of the COVID-19 pandemic. We have improved our manuscript by responding to your comments, as detailed below:

⇒We have addressed PLOS ONE's style requirements, including those for file naming.

2. Thank you for including your ethics statement: "The Research Ethics Boards of Health, Ministry of Health of the Royal Government of Bhutan (No. REBH/Approval/2018/089), and the Kyoto University Hospital and Graduate School of Medicine, Ethics Committee (No. R1796) approved the study protocol.". 

Please provide additional details regarding participant consent. In the ethics statement in the Methods and online submission information, please ensure that you have specified (1) whether consent was informed and (2) what type you obtained (for instance, written or verbal, and if verbal, how it was documented and witnessed). If your study included minors, state whether you obtained consent from parents or guardians. If the need for consent was waived by the ethics committee, please include this information.

⇒We have added an explanation of the ethical considerations of our study, as detailed below:

→(p12, line210-215 )　The Research Ethics Boards of Health, Ministry of Health of the Royal Government of Bhutan (No. REBH/Approval/2018/089) and the Ethics Committee of the Kyoto University Hospital and Graduate School of Medicine (No. R1796) approved the study protocol. The above ethics committees waived the requirement for individual informed consent as this secondary analysis used only de-identified data. We have submitted a written pledge of confidentiality to the Ministry of Health of the Royal Government of Bhutan. 

3. In your Methods section, please provide additional information about the participant recruitment method and the demographic details of your participants. Please ensure you have provided sufficient details to replicate the analyses such as:

- the recruitment date range (month and year)

⇒Thank you for your comments. The survey “National Survey for Noncommunicable Disease Risk Factors and Mental Health using WHO STEPS Approach in Bhutan – 2014” was conducted from March to June in 2014.　

→We added “from March to June”. (p 7, line 127)

- a description of any inclusion/exclusion criteria that were applied to participant recruitment

⇒Thank you for your comments.

In the “National Survey for Noncommunicable Disease Risk Factors and Mental Health using WHO STEPS Approach in Bhutan – 2014”, age 18 to 69 years was an inclusion criterion, and missing data were excluded from the analysis. Furthermore, in this study, pregnant women were excluded from the analysis because of health behavioral and biological differences. 

We have rephrased this as shown below:

→(p 11,　lines 202–204)　In the study, individuals aged 18 to 69 years were included. Those with missing data and pregnant women were excluded from the analysis because of health behavioral and biological differences.　 

- a statement as to whether your sample can be considered representative of a larger population

⇒Thank you for your comments.

We have added the following phrase:　

→(p 11, line 198-197)　 “to ensure they were representative of the national population”

- a description of how participants were recruited

- descriptions of where participants were recruited and where the research took place."

⇒Thank you for your comments. We have added the following sentences:

→(p 8, lines 134–135)　“According to the sampling plan, trained health staff recruited participants in each sampling field where the participants were living.” 

Further information about survey procedure can be obtained from the report “National Survey for Noncommunicable Disease Risk Factors and Mental Health Using WHO STEPS Approach in Bhutan – 2014”. We have provided supporting information quoted from the report (S1 File). 

4. Please provide a sample size and power calculation in the Methods, or discuss the reasons for not performing one before study initiation.

⇒Thank you for your comments. The survey “National Survey for Noncommunicable Disease Risk Factors and Mental Health Using WHO STEPS Approach in Bhutan – 2014” already considered the sample size and power calculation. Hence, we did not mention it in this paper; however, additional information from the Method part quoted from the report (S1 file) is now provided as supporting information.

Sample size and power calculation

“The Sample size estimate of the number of households to be surveyed with 95% confidence was calculated. The prevalence of overweight and obesity was 52.8% from the last STEPS survey carried out in Thimphu which was the closest value to 50%. d = margin of error. The expected half width of the confidence interval was taken as 0.05 for this study

　N=[1.96*1.96{0.528（1-0.528）}]/0.05*0.05 

Four domains were chosen based on men and women and two age groups: younger (18–39 years) and older (40–69 years), providing four age/sex estimates. Taking into account the number of domains and ensuring enough representation by either age-sex groups or urban-rural groups in men and women, and with a design effect of 1.5 to address the issue of cluster sampling, the expected sample size was as follows:

n = 382.9552 * 1.5 * 4 = 2297.7316

Assuming an expected 80% response rate, the final required sample size was 2912.

n = 2297.7316/0.8 = 2872.1646~ (rounded to 2912 for logistical ease)

⇒Thanks so much for your point out. We have addressed. 

→(p26,　Line438-449)　Our research was supported by　

1) Hiromi Kohori Segawa, K. MATSUSHITA FOUNDATION, Grant Number 19-048 http://matsushita-konosuke-zaidan.or.jp/en/

2) Yuichi Imanaka, Program of Collaborative Research at the Centre for Southeast Asian Studies (IPCR-CSEAS), Kyoto University 2018 type4 https://ipcr.cseas.kyoto-u.ac.jp/

3) Hiromi Kohori Segawa, Program of Collaborative Research at the Centre for Southeast Asian Studies (IPCR-CSEAS), Kyoto University 2020 type7 https://ipcr.cseas.kyoto-u.ac.jp/

4) Yuichi Imanaka, JSPS KAKENHI Grant Number JP19H01075 from the Japan Society for the Promotion of Science https://www.jsps.go.jp/english/

The funders played no role in the study design, data collection and analysis, decision to publish, or preparation of the manuscript.

6. Thank you for stating the following in your Competing Interests section: 

“NO”

⇒Thank you for your comments. We have mentioned this in our cover letter with the following information.

→The authors have declared that no competing interests exist. The funders played no role in the study design, data collection and analysis, decision to publish, or preparation of the manuscript.

7. We note that you have indicated that data from this study are available upon request. PLOS only allows data to be available upon request if there are legal or ethical restrictions on sharing data publicly. For more information on unacceptable data access restrictions, please see http://journals.plos.org/plosone/s/data-availability#loc-unacceptable-data-access-restrictions.

⇒Thank you for your comments. We have rephrased this information as shown below.

→(p 26,　lines 429–436)　Since the Ministry of Health, Bhutan is the authority for the data, the authors were not able to make it available in this manuscript. However, the original report (World Health Organization and Ministry of Health in Bhutan. National Survey for Noncommunicable Disease Risk Factors and Mental Health using WHO STEPS Approach in Bhutan – 2014) is available at https://apps.who.int/iris/handle/10665/204659. Requests to access the raw data can be directed to the Research Ethics Board of Health, Ministry of Health, Bhutan, or the Royal Government of Bhutan.

Additional Editor Comments (if provided):

Reviewers' comments:

Reviewer's Responses to Questions

Comments to the Author

1. Is the manuscript technically sound, and do the data support the conclusions?

Reviewer #1: No　　　　　　Reviewer #2: Yes　　Reviewer #3: Partly

 2. Has the statistical analysis been performed appropriately and rigorously?

 Reviewer #1: No　　Reviewer #2: Yes　　Reviewer #3: I Don't Know

 3. Have the authors made all data underlying the findings in their manuscript fully available?

 Reviewer #1: No　　Reviewer #2: No　　Reviewer #3: Yes

 4. Is the manuscript presented in an intelligible fashion and written in standard English?

 Reviewer #1: No　　Reviewer #2: Yes　　Reviewer #3: Yes

 5. Review Comments to the Author

 Reviewer #1: The major problems is the methodology and analysis done did not fit the study objectives. The study objective is to determine the factors associated with blood pressure measurement. It is not clear in the method the operational definition of blood pressure measurement in this study. What is the main outcome in this study? Is it behaviour practice of BP measurement for the past? duration or ever checked BP?

⇒Thank you for these important comments. We acknowledge the limitation of our explanation. We would like to make it clearer.

During the survey “National Survey for Noncommunicable Disease Risk Factors and Mental Health using WHO STEPS Approach in Bhutan – 2014”, blood pressure was checked 3 times by health workers. We could identify the gap between self-reported hypertension and examination based on the survey time (including diagnosis by health staff before the survey or current medication treatment). We believe that this gap is critical because individuals who have hypertension but are unaware should start to control their blood pressure. In particular, those who answered that they had no experience with blood pressure monitoring should be given attention. Periodic blood pressure monitoring is very important. As public health workers prevent worsening hypertension, we need to pay adequate attention to those with hypertension who did not have their blood pressure monitored before the survey. Hence, we wanted to identify what factors prevented them from being aware of their hypertension status before the survey. We wanted to identify whether they had concerns about their health but were unable have their blood pressure monitored or were unaware of their risk.

In the current study, we first wanted to identify factors related to experience with blood pressure monitoring before the survey. Second, we wanted to explore what factors are related to experience with blood pressure measurement among individuals with hypertension before the survey before the survey. 

To make these factors clear, we have made the following changes:

1) We have added second objective “To explore what factors are related to experience with blood pressure measurement among individuals with hypertension before the survey” (p 6,　lines 110–111 )　

2) We have added information about the time of diagnosis of hypertension (during or before the survey).

3) We have improved the explanation of the definition of hypertension used in this study.

→(p 9-10,　line 167-170 )　 Hypertension was defined by the following criteria: (1) systolic blood pressure (SBP) ≥ 140 mmHg or diastolic blood pressure (DBP) ≥ 90 mmHg measured as the average of three measurements taken by health staff in the survey; (2) a previous diagnosis of hypertension by healthcare workers; (3) current treatment with medication for hypertension

Reviewer #2: Reviewer Comments

Kohori-Segawa et al conducted a study to determine what characteristics are associated in Bhutanese people with the experience of having a blood pressure measurement. I find the article very interesting and neatly conducting. I have a few questions regarding the design:

1) In line x you mention patients with hypertension that had not had blood pressure measured, can you specify then how was the diagnosis made? Do you have any information regarding this? You mention this again in the conclusions in which you say people with hypertension have never had blood pressure measurements in their life so I do not understand how they know their hypertension status. Is it possible that they do not remember having their blood pressure measured rather than the measurement not being performed?

⇒Thank you for your comments. We acknowledge the shortcomings in the explanation. We would like to clarify. (Your comments are similar to the comments from reviewer 1; therefore, our responses are repeated.)

During the survey “National Survey for Noncommunicable Disease Risk Factors and Mental Health using WHO STEPS Approach in Bhutan – 2014”, blood pressure was checked 3 times by health workers. We could identify the gap between self-reported hypertension and examination based on the survey time (including diagnosis by health staff before the survey or current medication treatment). We believe that this gap is critical because individuals who have hypertension but are unaware should start to control their blood pressure. In particular, those who answered that they had no experience with blood pressure measurement should be given attention. Periodic blood pressure monitoring is very important. As public health workers prevent worsening hypertension, we need to pay adequate attention to those with hypertension who did not have their blood pressure measured before the survey. We wanted to identify whether they had concerns about their health but were unable have their blood pressure measured or were unaware of their risk.

To make these factors clear, we have made the following changes:

1) We have added second objective “To explore what factors are related to experience with blood pressure measurement among individuals with hypertension before the survey” (p 6,　lines 110–111 )　

2) We have added information about the time of diagnosis of hypertension (during or before the survey).

3) We have improved the explanation of the definition of hypertension used in this study.

→(p 9-10,　line 167-170 )　 Hypertension was defined by the following criteria: (1) systolic blood pressure (SBP) ≥ 140 mmHg or diastolic blood pressure (DBP) ≥ 90 mmHg measured as the average of three measurements taken by health staff in the survey; (2) a previous diagnosis of hypertension by healthcare workers; (3) current treatment with medication for hypertension

There is a possibility that the participants might not remember having their blood pressure measured as you mentioned. Such an unconscious measurement can be counted as not measured. If the participants forgot that they had their blood pressure measured, this indicates that they were not aware of the blood pressure measurement, so it is not a problem to count it as not having been taken. The important point is finding the gap in their practices, their wishes, and their real risk. 

2) Regarding multivariable analysis, I would like for you to specify adjustment variables rather than just listing the variable category (line 162)

⇒Thank you for your comments. We specified this point in the explanatory variables’ subsection (adjustment variables). To clarify, we added the explanation shown below and deleted the categories.

→(p 10,　line 177 )　 as mentioned in the explanatory variables.

3) In line 317 you mention than unemployed or self-employed men are less likely to have their blood pressure measured. A possible explanation could be the need to work on a daily basis to obtain income which makes it impossible to leave and seek for medical attention. I would like to know your thoughts about this.

⇒We acknowledge your valuable comments.

We have added the following explanation:

→(p 22–23,　lines 366–370 )　 It is possible that non-working or self-employed men must work on a daily basis to obtain income, making it impossible to leave work and seek medical attention, or they may not be able to seek medical attention without other interactions. Those who are employed may have received some health care through their employer or colleagues.

Finally I would suggest changing the wording in sentences mentioning previous studies.

⇒Thank you for your comments.　We have changed the wording about previous studies.

(p 20,　line 323 )　previous study →　A previous qualitative study

(p 20,　lines 324–325 )　this study →　this previous qualitative study

Reviewer #3:

1- Line 184: Can't find Figure1?. Does it refer to Table1?

⇒Thank you for your comments. It was Figure 1. PlosOne indicated to display the figures in different files not in the manuscript. Hence, we followed these directions.

2- It is difficult to understand the % in last column in table 1. How these % were calculated?

⇒Thank you for your comments. We calculated these data with the weight adjusted and adjusted for multistage cluster random sampling with SPSS. The total percentage is not 100 percent due to this adjustment. The last column is the point estimates after the adjustment. We did not include confidence intervals, which might have been confusing. Therefore, we have added 95% CIs and mentioned “estimates” instead of “adjusted” in Table 1. Furthermore, we have improved the footnotes in Table 1 and added S1 File.

(p 14 )　

1) Weight adjusted % (S1 File)

2) Population with blood pressure monitoring using the questionnaire item “Have you ever had your blood pressure measured by a doctor or other health worker?” Answers of “yes” were counted.

3) Estimated % adjusted with complex sampling: Population with blood pressure monitoring/Total target population (S1 File) 

We have added the explanation in the supporting information (S1 File). .

3-How to identify potential confounders? What are the potential confounders' variables to be included in multivariate analysis?

⇒We acknowledge your valuable comments. Health behavior such as blood pressure measurement is known to be affected by various factors. All of these various factors are associated with each other.

Kenneth et al (1988) proposed an ecological model for health promotion that focuses attention on both individual and social environmental factors as targets for health promotion interventions. It is important that “interpersonal” processes shape human behavior [1].

 Barton H et al (2006) also displayed the determinants of health and wellbeing in a human habitation map and mentioned “lifestyle” in various factors [2]. 

Reflecting on the Bhutanese situation, health policy is common for all Bhutanese across the country. Some but not all of the primary prevention programs for NCDs, such as displays of posters in health facilities, had already started in 2014 at the time of the survey. The number of primary health centers (PHCs) was also increasing in rural areas due to expansion of primary health care. Thus, we predicted that the situations of diagnosis and risk perception were also diverse. Furthermore, risk perception (threat) is also known to affect human behavior [3].

We selected individual and social environmental factors affecting interpersonal decisions or close networks, lifestyle-related factors, health conditions of hypertension, and family history of hypertension as biological risk perception factors.

1. McLeroy KR, Bibeau D, Steckler A, Glanz K. An Ecological Perspective on Health Promotion Programs. Health Education Quarterly. 1988;15(4):351-77.

2. Barton H, Grant M. A health map for the local human habitat. Journal of the Royal Society for the Promotion of Health. 2006;126(6):252-3.

3. Rosenstock IM, Strecher VJ, Becker MH. Social learning theory and the Health Belief Model. Health Educ Q. 1988;15(2):175-83.

According to the abovementioned theories, we selected the variables from “National survey for noncommunicable disease risk factors and mental health using WHO STEPS approach in Bhutan – 2014”.

We selected that gender and marital status; age, level of education, working status, residential area (urbanicity), income, and survey language Tobacco use, alcohol consumption, fruits and vegetables consumption, physical activity, estimated salt intake, having hypertension, and family history of hypertension. We put all these variables we selected to our models forcefully, after we checked the interaction effect and coefficient, excluding the residential area (urbanicity) due to adjusting cluster random sampling. 

We improved our explanation on the statistical methods and deleted the categories in explanatory 

4-I am not English native speaker, but I just feel that there might be a need to further improvement in English writing fashion?

⇒Thank you for your comments. English is our second language. We have asked a professional English copyeditor to review and edit our manuscript to ensure that it is written in standard English.

5-Title: It is too long; it should be revised?

⇒Thank you for your comments. We have shortened our title as below.

Old Title: Factors related to blood pressure measurement, with attention needed for unemployed men, poor women, and people’s health behaviors: A cross-sectional study in Bhutan

Revised title: Social and behavioral factors related to blood pressure measurement: A cross-sectional study in Bhutan

6-Income: Readers who are not from the Kingdom of Bhutan may not understand like "Nu.0–9,000; ...." Is it possible to convert into a commonly known like UDS?

⇒We acknowledge your valuable comments. We converted from Bhutanese Ngultrum (Nu) into USD.

→(p 9,　lines 145–147 )　

1 USD = 74.4707 BTN (Accessed on 11th August 2021 https://www.xe.com/)

Nu. 0–9,000　→0–120.8 USD

Nu. 9,001–30,000　　→120.9–402.9 USD

Nu. 30,001–60,000　　→403.0–805.7 USD

Nu. 60,001+　　→805.8+ USD

7-Discussion: Authors raised about "Considering barriers......" (line 284-285) and "Contradictions in health behavior...." (line 320), these are all the factors associated with blood pressure measure. The words .... barriers, contradictions might not appropriate to be used here?

⇒Thank you for your comments. We reconsidered appropriate subtitles for the discussion. We agree with your suggestion that the barriers or contradictions should be discussed in phase but are not needed in the title. We have now changed the subtitles in the discussion.

→(p 21,　line 334)　Considering barriers to self-employed or unemployed men, and poor women →Self-employed or non-working men and low income women

→(p 23,　line 373)　Contradictions in health behavior and risk perception → Health behavior and risk perception

 6. PLOS authors have the option to publish the peer review history of their article (what does this mean?). If published, this will include your full peer review and any attached files.

Do you want your identity to be public for this peer review? For information about this choice, including consent withdrawal, please see our Privacy Policy.

 Reviewer #1: No　Reviewer #2: Yes: Mercedes Aguilar-Soto　　Reviewer #3: No　Author：Yes

---

## [Decision Letter · Decision Letter 1]

19 Oct 2021

PONE-D-21-12027R1Social and behavioral factors related to blood pressure measurement: A cross-sectional study in BhutanPLOS ONE

Dear Dr. Imanaka,

Thank you for submitting your manuscript to PLOS ONE. After careful consideration, we feel that it has merit but does not fully meet PLOS ONE’s publication criteria as it currently stands. Therefore, we invite you to submit a revised version of the manuscript that addresses the points raised during the review process.

ACADEMIC EDITOR: 

The manuscript is interesting but will require minor revisions.<o:p></o:p>

While they recognize the potential interest of the subject studied, the Reviewers raised some concerns that need to be properly addressed.<o:p></o:p>

We look forward to receiving your revised manuscript.

Kind regards,

Marcelo Arruda Nakazone, M.D., Ph.D.

Academic Editor

PLOS ONE

Journal Requirements:

Reviewers' comments:

Reviewer's Responses to Questions

**Comments to the Author**

1. If the authors have adequately addressed your comments raised in a previous round of review and you feel that this manuscript is now acceptable for publication, you may indicate that here to bypass the “Comments to the Author” section, enter your conflict of interest statement in the “Confidential to Editor” section, and submit your "Accept" recommendation.

Reviewer #1: All comments have been addressed

Reviewer #2: All comments have been addressed

Reviewer #3: All comments have been addressed

2. Is the manuscript technically sound, and do the data support the conclusions?

Reviewer #1: Yes

Reviewer #2: Partly

Reviewer #3: Yes

3. Has the statistical analysis been performed appropriately and rigorously? 

Reviewer #1: Yes

Reviewer #2: No

Reviewer #3: I Don't Know

4. Have the authors made all data underlying the findings in their manuscript fully available?

Reviewer #1: Yes

Reviewer #2: Yes

Reviewer #3: Yes

5. Is the manuscript presented in an intelligible fashion and written in standard English?

Reviewer #1: Yes

Reviewer #2: Yes

Reviewer #3: Yes

6. Review Comments to the Author

Reviewer #1: The author has adequately addressed all comments based form previous reviewers and the manuscript seem to be sound. However, I would like to advise the author to add the p value for the multivariate or bivariate analysis and highlight- bold the results that are significant. This can help the readers to recognized those significant factors in the table easily.

Reviewer #2: Yuichi Imanaka et al conducted a study to determine social and behavioral factors that determine having blood pressure measurements in Bhutan. They have corrected prior comments but there are some changes that are still needed.

Line 55: had a lower likelihood of undergoing blood pressure measurement

Correct wording.

I would further specify everytime you mention hypertensive patients did not have their blood pressure measured that this was after the first diagnosis measurement. (Line 65, 66)

LINE 88 : Have showed to be necessary instead of has

Again in line 94 you mention people with hypertension have not had their blood pressure measured but this is impossible. Please address this like you did previously. In line 96 you mention according to NCD survey reports, “Around 96 (31.3%) of the study population had never had their blood pressure checkup”, but this again does not make sense if they already have the diagnosis. They needed at least 2 measurements performed priorly. Perhaps you should clarify “follow-up” blood pressure in all sentences regarding new measurements.

Line 163: How was salt intake estimated? Based on a diet questionnaire?

Line 170: I do not understand why you used family history on hypertension as a definition for diagnosis. I suggest you remove this definition.

Line 176: I do not understand what you mean by this: All of the data are available and considered to be related to the dependent variable from the previous literature as mentioned in the explanatory variables.

I would suggest you specify very clearly which patients were included: only patients who had prior diagnosis of hypertension defined as known diagnosis or those who were diagnosed during the National Survey for Noncommunicable Disease Risk Factors and Mental Health.

Line 205: Again it is not clear why you made sub-analysis for people with hypertension, you need to be very careful with wording as to say that your objective is to know factors that determine the experiences of having blood pressure measurements and whether they differ for patients who had prior hypertension diagnosis and those who did not.

In the results section, if variables have a parametric distribution, it is better to state mean and standard deviation rather than standard error.

In table 1 rename variable Gender_Marital Status since it looks like you just left the statistical program output without correcting variable names.

In table two remove the parenthesis after 1 and 2 super indexes. Income variable in table 2 is not in the same format as the rest of the table.

In line 317 you still insist that patients with hypertension never underwent blood pressure measurement in their lives, but this, again, is not possible since that is the only way to diagnose hypertension is measuring blood pressure. You have explained in previous responses what you mean but you need to be very explicit about it throughout your discussion.

Check wording in line 352.

Reviewer #3: I have no idea whether data analysis was done appropriately and rigorously. However, the presentation of the results is clear and response to the research objectives.

7. PLOS authors have the option to publish the peer review history of their article (what does this mean?). If published, this will include your full peer review and any attached files.

Reviewer #1: No

Reviewer #2: **Yes: **Mercedes Aguilar-Soto

Reviewer #3: No

---

## [Author Response · Author response to Decision Letter 1]

31 Mar 2022

Dear Editor and reviewers,

We really appreciate your review and feedback during these difficult times of the COVID-19 pandemic. We took long time to learn causal inference and modify. However, now we have improved our manuscript by responding to your comments, as detailed below:

Reviewer #1: The author has adequately addressed all comments-based form previous reviewers and the manuscript seem to be sound. However, I would like to advise the author to add the p value for the multivariate or bivariate analysis and highlight- bold the results that are significant. This can help the readers to recognized those significant factors in the table easily.

⇒Thank you so much for your advice. I added p-value as much as possible and made them bold for significant results. However, in complex-sampling module Ver23, we are not able to calculate p-value individually in our logistic multivariable regression model. I asked IBM company, they guided us if we want to get p-value individually we need to change all references in our data set. 

Reviewer #2: Yuichi Imanaka et al conducted a study to determine social and behavioral factors that determine having blood pressure measurements in Bhutan. They have corrected prior comments but there are some changes that are still needed.

Line 55: had a lower likelihood of undergoing blood pressure measurement

Correct wording.

⇒Thank you for your comments. However, we think the wording are correct. We think that likelihood Is better than odds in this case due to complex survey module, and proofed by professional English editor. In case of “undergoing “makes confuse, we changed the word from” undergoing” to “having experience”. (Line59, 62)

I would further specify everytime you mention hypertensive patients did not have their blood pressure measured that this was after the first diagnosis measurement. (Line 65, 66)

⇒Thank you for your comments. We added “before the STEPS survey” and modified explanation. (Line 67-68, 96,104-106, 114-116, 173,289,365)

LINE 88 : Have showed to be necessary instead of has

⇒Thank you for your advice.　We have changed from “have” to “has”.(Line90)

Again in line 94 you mention people with hypertension have not had their blood pressure measured but this is impossible. Please address this like you did previously. In line 96 you mention according to NCD survey reports, “Around 96 (31.3%) of the study population had never had their blood pressure checkup”, but this again does not make sense if they already have the diagnosis. They needed at least 2 measurements performed priorly. Perhaps you should clarify “follow-up” blood pressure in all sentences regarding new measurements.

⇒Thank you for your comments. I added explanation “before the previous field work conducted in 2017”. However, we can’t modify the quoted sentences from previous government NCD survey report. We mention whom have no experience of high blood measurement before we met or before survey. During field work or the survey, we diagnosed their hypertension. (Line96)

Line 163: How was salt intake estimated? Based on a diet questionnaire?

⇒Thank you for your comments. We estimated salt intake based on spot urine. Details of the method to estimate salt intake are now included in the manuscript. 

Salt intake was calculated using the Tanaka Formula (Tanaka T, Okamura T, Miura K, Kadowaki T, Ueshima H, Nakagawa H, et al. A simple method to estimate populational 24-h urinary sodium and potassium excretion using a casual urine specimen. Journal of Human Hypertension. 2002;16(2):97-103).

Salt Intake per day(g) = (21.98 * ((Spot Urinary Sodium/

(Spot Urinary Creatinine *10/0.0884)) * (((14.89* Weight (Kg))+(16.14*height(m))*(2.04*age))-2244.45))**0.392)/17.1 (Line167-169)

Line 170: I do not understand why you used family history on hypertension as a definition for diagnosis. I suggest you remove this definition.

⇒Thank you for your suggestion. We have removed it. (Line175-176)

Line 176: I do not understand what you mean by this: All of the data are available and considered to be related to the dependent variable from the previous literature as mentioned in the explanatory variables.

⇒Thank you for your comments. We have improved the sentence as below

All available and selected data from the STEPS survey are considered related to the objective variable from previous literature. Among those variables that may be related to blood pressure measurement, we selected the explanatory variables to be fed into the model using the Directed Acyclic Graph (S3 File). (Line180-183)

I would suggest you specify very clearly which patients were included: only patients who had prior diagnosis of hypertension defined as known diagnosis or those who were diagnosed during the National Survey for Noncommunicable Disease Risk Factors and Mental Health.

⇒Thank you for your suggestion. Our explanation may make you confuse. We include all aged 18 to 69 years. We didn’t exclude normal blood pressure(Figure1). We just evaluate the hypertension during NCD STEPS survey. We added the explanation of the definition about hypertension through our paper. 

Line 205: Again, it is not clear why you made sub-analysis for people with hypertension, you need to be very careful with wording as to say that your objective is to know factors that determine the experiences of having blood pressure measurements and whether they differ for patients who had prior hypertension diagnosis and those who did not.

⇒Thank you for your meaningful advice.　We deeply considered and erased the subgroup analysis for multivariable regression models.”

In the results section, if variables have a parametric distribution, it is better to state mean and standard deviation rather than standard error.

⇒Thank you for your advice. We have added standard deviations in Table1. We recognize that logistic regression does not matter whether variables have a parametric distribution or not. And we adjusted the weight and cluster using SPSS complex sampling module, so it is better to use Standard Error than Standard Deviation. So, we kept standard error too. (Table1,Line,206-207,231)

In table 1 rename variable Gender_Marital Status since it looks like you just left the statistical program output without correcting variable names.

⇒Thank you for your advice. We deeply considered and changed not only names but also our models. We divided the combined variables as original. Because we can show the effects individually. (Line153-154, S3File)

In table two remove the parenthesis after 1 and 2 super indexes. Income variable in table 2 is not in the same format as the rest of the table.

⇒Thank you for your advice. We improved it.

In line 317 you still insist that patients with hypertension never underwent blood pressure measurement in their lives, but this, again, is not possible since that is the only way to diagnose hypertension is measuring blood pressure. You have explained in previous responses what you mean but you need to be very explicit about it throughout your discussion.

⇒Thank you for your meaningful advice. 

We have added the explanation throughout our discussion. All descriptions include the definition of Hypertension in this paper so that reader can understand who has never had a blood pressure measurement prior to the NCD STEPS survey. Hypertension was determined during the NCD STEPS survey, and the experience of blood pressure measurement was prior to the NCD STEPS survey. (Line 67-68, 96,104-106, 114-116, 173,289,365) 

From this your advice, we also learned about causal inferences in the field of epidemiology, considered that there is a clear time difference between the blood pressure measurement experience and the hypertension, and reconsidered our multivariate analysis model. The variable of hypertension should be collider in our models (S3 File). 

Check wording in line 352. 

⇒Thank you for advice. We have changed as below

Lower income women might also experience some barriers to visit the hospital, as previous studies in other countries have indicate. (Line324-325)

 Reviewer #3: I have no idea whether data analysis was done appropriately and rigorously. However, the presentation of the results is clear and response to the research objectives.

⇒Thank you so much for your comments. Regarding the data analysis, we have done the previous revisions rigorously. However, we have reconsidered and changed models from view point of causal inference. There was no significant change in the results from the last revision. 

Sincerely yours

---

## [Decision Letter · Decision Letter 2]

4 May 2022

PONE-D-21-12027R2Social and behavioral factors related to blood pressure measurement: A cross-sectional study in BhutanPLOS ONE

Dear Dr. Imanaka,

Thank you for submitting your manuscript to PLOS ONE. After careful consideration, we feel that it has merit but does not fully meet PLOS ONE’s publication criteria as it currently stands. Therefore, we invite you to submit a revised version of the manuscript that addresses the points raised during the review process.

ACADEMIC EDITOR:

Previous comments have been addressed. But the manuscript will require a minor revision.<o:p></o:p>

While they recognize the potential interest of the subject studied, one of the Reviewers raised additional comments that need to be properly addressed.<o:p></o:p>

We look forward to receiving your revised manuscript.

Kind regards,

Marcelo Arruda Nakazone, M.D., Ph.D.

Academic Editor

PLOS ONE

Journal Requirements:

Reviewers' comments:

Reviewer's Responses to Questions

**Comments to the Author**

1. If the authors have adequately addressed your comments raised in a previous round of review and you feel that this manuscript is now acceptable for publication, you may indicate that here to bypass the “Comments to the Author” section, enter your conflict of interest statement in the “Confidential to Editor” section, and submit your "Accept" recommendation.

Reviewer #2: (No Response)

2. Is the manuscript technically sound, and do the data support the conclusions?

Reviewer #2: Yes

3. Has the statistical analysis been performed appropriately and rigorously? 

Reviewer #2: Yes

4. Have the authors made all data underlying the findings in their manuscript fully available?

Reviewer #2: No

5. Is the manuscript presented in an intelligible fashion and written in standard English?

Reviewer #2: No

6. Review Comments to the Author

Reviewer #2: Kohori-Segawa et al conducted a study to determine what characteristics are associated in Bhutanese people with the experience of having a blood pressure measurement. The article is interesting, however I still have a few corrections that need to be revised:

1) In line 174 the final definition of hypertension is having medication for hypertension and a positive family history of hypertension. I would suggest citing where was this definition taken from or how do you justify it.

2) Line 199 repeats again in line 201.

3) In line 203 I would further clarify which variables you identified as confounders in your multivariable analyses.

4) Line 212-216 are written incorrectly.

5) Line 259 is missing reference.

6) In line 266 where you say you did not find an association between Q3 and the experience of blood pressure measurement, why are you focusing only on Q3? If it is because it is the only one with a statistical association you should write it like that. And why are highest income quartiles categorized as Q1? Usually Q1 is the lowest quartile.

7) Line 292 is repetitive using diseases twice.

8) Line 306 background should be written as one word.

9) Line 306 this subtitle does not accurately reflect what you explain in the paragraph.

10) In line 324 you mention other barriers but not explain further which ones.

11) Line 338 should say “have more opportunities on having their health examined” rather than the actual wording.

12) In line 347 I would further explain your hypotheses on this association or move line 352 next to this paragraph.

13) Line 350 is repetitive, I would suggest: Those with a family history of hypertension or those with hypertension had more experiences with blood pressure measurement in all analyses probably because they have an increased awareness of the health threats they might face.

Finally, there are still several grammar mistakes that I would suggest you proofread.

7. PLOS authors have the option to publish the peer review history of their article (what does this mean?). If published, this will include your full peer review and any attached files.

Reviewer #2: **Yes: **Mercedes Aguilar-Soto

---

## [Author Response · Author response to Decision Letter 2]

20 May 2022

We appreciate your review and feedback.

We have addressed all of the reviewers’ comments as below

1) In line 174 the final definition of hypertension is having medication for hypertension and a positive family history of hypertension. I would suggest citing where was this definition taken from or how do you justify it.

⇒Thank you for your suggestion. We realized that the definition was mistakenly written as “and a positive family history of hypertension”. Therefore, we removed it. 

2) Line 199 repeats again in line 201.

⇒Thank you for your confirmation. Starting words were the same as “we evaluated…”. Actually, there is not repetition in the contents. We changed the sentence to better the understanding. 

Line 199: We evaluated the association between the experiences of having blood pressure measurements and individual and social environmental factors affecting interpersonal decisions or close networks through descriptive and univariable logistic regression, the association between the experiences of having blood pressure measurements and high-risk behaviors of hypertension factors through descriptive and multivariable logistic regression.

3) In line 203 I would further clarify which variables you identified as confounders in your multivariable analyses.

⇒Thank you for your advice. We added the explanation as below.

Line203-206: Through Directed Acyclic Graph, we defined minimal sufficient adjustment sets for estimating the total effect of high-risk behaviors of hypertension factors on the experiences of having blood pressure measurements.

4) Line 212-216 are written incorrectly.

⇒Thank you reviewer. We modified the sentence as below.

Line214: In the study, participants aged 18 to 69 years were included. Variables with missing data and pregnant women were excluded from the analysis because of health behavioral and biological differences.

5) Line 259 is missing reference.

⇒Thank you reviewer for this notification.

Actually, the reference here relates to reference of Odds calculated (i.e. no formal education). We changed from (Ref) to (OR:1).

6) In line 266 where you say you did not find an association between Q3 and the experience of blood pressure measurement, why are you focusing only on Q3? If it is because it is the only one with a statistical association you should write it like that. And why are highest income quartiles categorized as Q1? Usually Q1 is the lowest quartile.

⇒Thank you reviewer for this critical comment and the valuable suggestion. We modified the sentence as below.

Line263: In terms of income, all participants in the Q4 (Nu. 60,001+: 805.8+ USD (OR:2.404, 95% CI:1.575 –3.667)) and Q3 category (Nu 30,001–60,000: 403.0–805.7 USD (OR:2.546, 95% CI: 1.640–3.955)) had an increased likelihood of having experienced blood pressure measurement compared with those in the lowest income category Q1 (Nu 0–9,000: 0–120.8 USD[OR:1]). We could find the significant association between Q1, Q3, Q4 and the experience of blood pressure measurement, but we could not find a significant association between Q2 (Nu 9,001–30,000: 120.9–402.9 USD) and the experience of blood pressure measurement.

7) Line 292 is repetitive using diseases twice.

⇒ “cardiovascular diseases” is removed from the sentence

Line 294: Early detection and treatment of hypertension can be effective in preventing other diseases [3,17]

8) Line 306 background should be written as one word.

⇒Thank you. “Back ground” is changed to single word “background” as suggested

9) Line 306 this subtitle does not accurately reflect what you explain in the paragraph.

⇒Thank you for this critique. We modified from “Social background factors and gender difference: Self-employed, non-working or low-income men, and low-income women” to “Social background factors and gender differences”.

10) In line 324 you mention other barriers but not explain further which ones.

We added the explanation as below.

Line326-328: Lower income women might also experience some barriers to visiting the hospital, as previous studies in other countries have indicated, such as insufficient education, gender problems, insufficient family supports, etc. [42].

11) Line 338 should say “have more opportunities on having their health examined” rather than the actual wording.

⇒Thank you reviewer for your suggestion. We modified the sentence as below.

Line341: Those who are employed may have more opportunities on having their health examined and other services from their companies than those who are self-employed or non-working [47].

12) In line 347 I would further explain your hypotheses on this association or move line 352 next to this paragraph.

⇒We moved the sentences as suggested.

13) Line 350 is repetitive, I would suggest: Those with a family history of hypertension or those with hypertension had more experiences with blood pressure measurement in all analyses probably because they have an increased awareness of the health threats they might face.

⇒Thank you reviewer for your suggestion. Much appreciated. We improved the sentences as below.

Line350-356: Those with a family history of hypertension or those with hypertension had more experiences with blood pressure measurement in all analyses probably because they have an increased awareness of the health threats they might face. People with unhealthy lifestyle may not be aware of health threats or perceived health vulnerability, and one of the characteristics of people with healthy lifestyle may be the ability to perceive health threats, regardless of whether they have pre-existing health problems or not. People with unhealthy lifestyle (high salt intake or tobacco use) had less experience with blood pressure measurement.

Finally, there are still several grammar mistakes that I would suggest you proofread.

We agree and therefore, we proofread again.

Sincerely yours

---

## [Decision Letter · Decision Letter 3]

6 Jun 2022

PONE-D-21-12027R3Social and behavioral factors related to blood pressure measurement: A cross-sectional study in BhutanPLOS ONE

Dear Dr. Imanaka,

Thank you for submitting your manuscript to PLOS ONE. After careful consideration, we feel that it has merit but does not fully meet PLOS ONE’s publication criteria as it currently stands. Therefore, we invite you to submit a revised version of the manuscript that addresses the points raised during the review process.

Previous comments have been addressed. However, the manuscript will require a minor revision.

Reviewer #2 still raised some points that require the authors attention before publication.

In this context, I strongly recommend an English speaker proofreading since grammar mistakes remain persistent through the review process.

We look forward to receiving your revised manuscript.

Kind regards,

Marcelo Arruda Nakazone, M.D., Ph.D.

Academic Editor

PLOS ONE

Journal Requirements:

Reviewers' comments:

Reviewer's Responses to Questions

**Comments to the Author**

1. If the authors have adequately addressed your comments raised in a previous round of review and you feel that this manuscript is now acceptable for publication, you may indicate that here to bypass the “Comments to the Author” section, enter your conflict of interest statement in the “Confidential to Editor” section, and submit your "Accept" recommendation.

Reviewer #2: All comments have been addressed

2. Is the manuscript technically sound, and do the data support the conclusions?

Reviewer #2: Yes

3. Has the statistical analysis been performed appropriately and rigorously? 

Reviewer #2: Yes

4. Have the authors made all data underlying the findings in their manuscript fully available?

Reviewer #2: No

5. Is the manuscript presented in an intelligible fashion and written in standard English?

Reviewer #2: No

6. Review Comments to the Author

Reviewer #2: In the abstract: Cardiovascular disease is a leading cause of death in the Kingdom of Bhutan, and the early detection of hypertension is critical for preventing it.

Again in the abstract clarify that “26% of the participants with hypertension never underwent blood pressure measurements” after their diagnosis since it is imposible to diagnose hypertension without a blood measurement, or whether this patients were identified during the STEPS survey.

Line 59 says “having experienced with blood pressure measurement.” Check grammar.

Check grammar in lines 80-81, 215-216, 242, 243, 245, 253, 278, 300, 336.

Line 238 would be better this way: Among those with hypertension, 26.3% (crude n=197, SE: 2.5%, 95% CI :21.7–31.4%) had never had blood pressure measurement before.

In line 268 you say Q1 was also significantly associated with the experience of blood pressure measurment, however Q1 was the reference so I do not understand this line.

I highly recommend an English speaker proofreading since grammar mistakes have been present through revisions.

7. PLOS authors have the option to publish the peer review history of their article (what does this mean?). If published, this will include your full peer review and any attached files.

Reviewer #2: **Yes: **Mercedes Aguilar-Soto

---

## [Author Response · Author response to Decision Letter 3]

20 Jun 2022

We really appreciate your review and feedback.

Reviewer #2: In the abstract: Cardiovascular disease is a leading cause of death in the Kingdom of Bhutan, and the early detection of hypertension is critical for preventing it.

Again in the abstract clarify that “26% of the participants with hypertension never underwent blood pressure measurements” after their diagnosis since it is imposible to diagnose hypertension without a blood measurement, or whether this patients were identified during the STEPS survey.

⇒Thank you for your comment. We asked the English proofreader and had improved the sentence as follows.　

Line53:“Approximately 26% of those with hypertension who were detected during the STEPS survey had never had their blood pressure measured.”

Line 59 says “having experienced with blood pressure measurement.” Check grammar.

⇒Thank you for your comment. We asked the English proofreader and had improved the sentence as follows.　

Line60: “A family history of hypertension (OR: 2.019, 95% CI: 1.549–2.243) increased the likelihood of having experienced a blood pressure measurement in both men and women. “

Check grammar in lines 80-81, 215-216, 242, 243, 245, 253, 278, 300, 336.

⇒Thank you for your comments.　

We have hired a professional English proofreader and improved some sentences.

The improved points were highlighted in green with you mentioning them, highlighted in yellow was without you mentioning them.

Line 238 would be better this way: Among those with hypertension, 26.3% (crude n=197, SE: 2.5%, 95% CI :21.7–31.4%) had never had blood pressure measurement before.

⇒Thank you for your advice. We had improved as follows.

Line257: “Among those with hypertension, 26.3% (crude n=197, SE: 2.5%, 95% CI :21.7–31.4%) had never had a blood pressure measurement before.

In line 268 you say Q1 was also significantly associated with the experience of blood pressure measurment, however Q1 was the reference so I do not understand this line.

We could find the significant association between Q3, Q4 and the experience of blood pressure measurement, but we could not find a significant association between Q2 (Nu 9,001–30,000: 120.9–402.9 USD) and the experience of blood pressure measurement.

⇒Thank you for your advice. We asked the English proofreader and had improved the sentence as follows.

 Line 285: “In terms of income, all participants in Quartile-4 (Nu. 60,001+: USD 805.8+ (OR: 2.404, 95% CI: 1.575 –3.667)) and Quartile-3 (Nu. 30,001–60,000: USD 403.0–805.7 (OR: 2.546, 95% CI: 1.640–3.955)) had an increased likelihood of previous blood pressure measurement compared with those in the lowest income category, Quartile-1 (Nu. 0–9,000: USD 0–120.8 [OR: 1.000]). We found a significant association between Quartile-3, Quartile-4, and previous blood pressure measurement but not between Q2 (Nu. 9,001–30,000: USD 120.9–402.9) and previous blood pressure measurement.”

I highly recommend an English speaker proofreading since grammar mistakes have been present through revisions.

⇒Thank you for your advice. We asked the English speaker professional proofreader and had improved the whole sentence.

---

## [Decision Letter · Decision Letter 4]

11 Jul 2022

Social and behavioral factors related to blood pressure measurement: A cross-sectional study in Bhutan

PONE-D-21-12027R4

Dear Dr. Imanaka,

We’re pleased to inform you that your manuscript has been judged scientifically suitable for publication and will be formally accepted for publication once it meets all outstanding technical requirements.

Kind regards,

Marcelo Arruda Nakazone, M.D., Ph.D.

Academic Editor

PLOS ONE

Additional Editor Comments (optional):

Reviewers' comments:

Reviewer's Responses to Questions

**Comments to the Author**

1. If the authors have adequately addressed your comments raised in a previous round of review and you feel that this manuscript is now acceptable for publication, you may indicate that here to bypass the “Comments to the Author” section, enter your conflict of interest statement in the “Confidential to Editor” section, and submit your "Accept" recommendation.

Reviewer #2: All comments have been addressed

2. Is the manuscript technically sound, and do the data support the conclusions?

Reviewer #2: Yes

3. Has the statistical analysis been performed appropriately and rigorously? 

Reviewer #2: Yes

4. Have the authors made all data underlying the findings in their manuscript fully available?

Reviewer #2: Yes

5. Is the manuscript presented in an intelligible fashion and written in standard English?

Reviewer #2: Yes

6. Review Comments to the Author

Reviewer #2: Imanaka et al conducted a study in Buthan in order to determine health behaviour attitudes and social factors that are associated with patients getting blood pressure measurements.

After several prior revissions all comments have been adressed, methodology has been explained better and grammar has been profread.

7. PLOS authors have the option to publish the peer review history of their article (what does this mean?). If published, this will include your full peer review and any attached files.

Reviewer #2: **Yes: **Mercedes Aguilar-Soto

---

## [Editor Report · Acceptance letter]

8 Aug 2022

PONE-D-21-12027R4 

Social and behavioral factors related to blood pressure measurement: A cross-sectional study in Bhutan 

Dear Dr. Imanaka:

I'm pleased to inform you that your manuscript has been deemed suitable for publication in PLOS ONE. Congratulations! Your manuscript is now with our production department. 

Kind regards, 

on behalf of

Professor Marcelo Arruda Nakazone 

Academic Editor

PLOS ONE